# Seg4Diff: Unveiling Open-Vocabulary Segmentation in Text-to-Image Diffusion Transformers

**Chaehyun Kim[1]**    **Heeseong Shin[1]**    **Eunbeen Hong[1]**    **Heeji Yoon[1]**
**Anurag Arnab**    **Paul Hongsuck Seo[2]**    **Sunghwan Hong[3,†]**    **Seungryong Kim[1,†]**
[1]KAIST AI    [2]Korea University    [3]ETH Zürich AI Center, CVG, PRS
https://cvlab-kaist.github.io/Seg4Diff

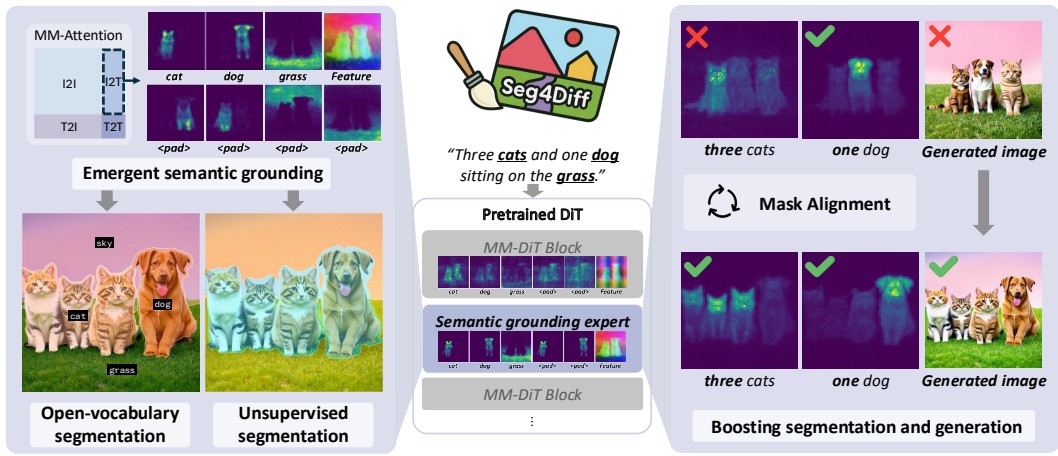

Figure 1: We introduce **Seg4Diff**, a systematic framework designed to ***analyze and enhance*** the emergent semantic grounding capabilities of multi-modal diffusion transformer (MM-DiT) blocks in text-to-image diffusion transformers (DiTs). Here, ***semantic grounding expert*** refers to a specific MM-DiT block responsible for establishing semantic alignment between text and image features.

## Abstract

Text-to-image diffusion models excel at translating language prompts into photore-alistic images by implicitly grounding textual concepts through their cross-modal attention mechanisms. Recent multi-modal diffusion transformers extend this by introducing joint self-attention over concatenated image and text tokens, enabling richer and more scalable cross-modal alignment. However, a detailed understanding of how and where these attention maps contribute to image generation remains limited. In this paper, we introduce **Seg4Diff** (**Seg**mentation **for Diff**usion), a systematic framework for analyzing the attention structures of MM-DiT, with a focus on how specific layers propagate semantic information from text to image. Through comprehensive analysis, we identify a *semantic grounding expert* layer, a specific MM-DiT block that consistently aligns text tokens with spatially co-herent image regions, naturally producing high-quality semantic segmentation masks. We further demonstrate that applying a lightweight fine-tuning scheme with mask-annotated image data enhances the semantic grouping capabilities of these layers and thereby improves both segmentation performance and generated image fidelity. Our findings demonstrate that semantic grouping is an emergent property of diffusion transformers and can be selectively amplified to advance both segmentation and generation performance, paving the way for unified models that bridge visual perception and generation.

39th Conference on Neural Information Processing Systems (NeurIPS 2025).

# 1 Introduction

The emergence of text-to-image (T2I) diffusion models have revolutionized visual content creation by allowing users to generate photorealistic images from natural-language prompts [46, 22, 52]. The success of these models imply that text-to-image diffusion models are highly capable of associating visual and textual representations, as the generated images accurately reflect the given textual prompt.

In this regard, a growing body of work shows that this semantic grounding is encoded in the cross-modal attention maps of the diffusion models [20, 41, 49, 43], which have already been actively exploited for generative downstream tasks such as image editing, inpainting and personalization [48, 20, 5, 39]. Furthermore, several studies [60, 54] show that the cross-attention maps can be used to perform visual perception tasks such as open-vocabulary semantic segmentation, revealing that diffusion models can double as training-free segmentation models. However, the attention map produced by the standard U-Net-based [47, 46] models is often noisy and spatially fragmented [6], limiting the fidelity of the resulting masks when leveraged without heuristic refinement.

Recent advances replace the U-Net architecture with diffusion transformers (DiTs) [42], which benefits from the strong representation power and scalability of transformers for achieving state-of-the-art image generation quality. In particular, the multi-modal diffusion transformer (MM-DiT) [15, 53, 3] introduces multi-modal attention, which concatenates text and image tokens and applies joint self-attention, enabling richer cross-modal interaction. This encourages us to explore the internal representations and the characteristics of the MM-DiT-based models. However, as opposed to U-Net based models, which have been extensively studied [38], DiT-based models and their characteristics remain underexplored, in which a detailed understanding of how and where these attention maps contribute to image generation remains limited.

In this paper, we first conduct an in-depth analysis of the joint attention mechanism in MM-DiT models. We characterize the distribution of attention scores to discover active cross-modal interaction, and complement this with feature similarity measure and norm analysis to assess which modality—textual or visual—exerts greater influence on the output representations. Together, these perspectives isolate a small subset of layers that consistently align textual semantics with contiguous image regions. These observations motivate us to further analyze this emergent semantic grounding capability.

Subsequently, we propose an open-vocabulary segmentation scheme leveraging MM-DiT, which demonstrates that the identified layer, ***semantic grounding expert***, yields high-quality segmentation masks, confirming their inherently competitive capability for open-vocabulary semantic segmentation. Extended analysis reveals that these attention map is in fact further decomposed into attention heads, which capture distinct parts of the semantic region and finally sum up to construct a complete semantic mask. Moreover, without explicit class information, we reveal that `<pad>` tokens can decompose the image into meaningful semantic regions, acting as anchors for the unconditional generation.

Building on these findings, we introduce a lightweight fine-tuning scheme, called ***mask alignment for segmentation and generation (MAGNET)***, that explicitly strengthens semantic grouping by leveraging mask-annotated image data in selected MM-DiT layers. Our primary motivation is to reinforce the emergent segmentation capability of these layers, enabling more accurate open-vocabulary semantic masks. Interestingly, this targeted adaptation also yields a modest but consistent improvement in image synthesis quality as a by-product of enhancing the cross-modal semantic alignment. Taken together, our stepwise analysis and selective refinement establish semantic grouping as a salient property of diffusion transformers for text-conditioned image generation—one that can be leveraged with minimal compute to advance both dense recognition and generative fidelity. These results illuminate the internal dynamics of diffusion transformers and point toward unified models that excel at both generation and perception.

In summary, our contributions are as follows:

- We provide in-depth analysis and investigation of learned representations of MM-DiT within its multi-modal attention layers.
- We identify critical layers in MM-DiT that are essential for preserving text-conditioned semantics throughout the generation process, which reveals strong semantic grounding.
- We demonstrate a zero-shot segmentation scheme to extract segmentation masks from the specific layers, and further enhance generation quality by enhancing their localization capability.

## 2 Related Work

**Emergent properties of diffusion models.**    Diffusion models, which generate images by iteratively denoising Gaussian noise, have transformed generative modeling [22, 52]. U-Net-based text-to-image models [46] surpassed GANs [18], while recent transformer backbones [42] further raised the bar. In particular, multi-modal diffusion transformer (MM-DiT) [15], achieve stronger scalability and semantic alignment by fusing image and text tokens through joint attention. Beyond generation, pretrained diffusion models have been adapted to perception tasks such as correspondence [67, 56], segmentation [59], tracking [37], depth estimation [29, 26] and video understanding [58]. On the other hand, inspired by Prompt-to-Prompt [20], which first proposed to leverage the cross-modal attention maps of text-to-image diffusion models revealing that text information is delicately aggregated into image generation via the attention mechanism, a rich line of works explored attention maps in different areas, such as image editing [7] and conditional generation [1]. Building on this, we focus specifically on analyzing the joint attention mechanism of MM-DiT, uncovering its rich semantic structure and its potential for segmentation. This perspective not only bridges generation and perception but also highlights attention as a central building block for semantic reasoning in diffusion models.

**Leveraging diffusion models for perception tasks.**    Recently, text-to-image diffusion models [46, 43, 15] have been adapted beyond generation to tackle discriminative vision tasks with remarkable success. For segmentation, ODISE [64] integrates a frozen diffusion backbone with CLIP, while adapter-based methods [68] leverage cross-attention for referring segmentation. For geometry, lightweight fine-tuning of diffusion model for depth or normal estimation [63, 17] enhances spatial detail and generalization. In correspondence, methods like DiffMatch [40] achieve robustness to textureless regions, while leveraging frozen features [56] still rival supervised baselines [9, 10, 24]. Collectively, these works show that diffusion models encode rich priors transferable to perception. However, most approaches treat diffusion feature as static representations, leaving the role of internal attention mechanism underexplored, particularly in architectures like MM-DiT.

**Diffusion models for segmentation.**    Diffusion models [4, 15] excel at photorealistic image synthesis by aligning semantic cues between images and text, which has spurred efforts to repurpose their learned representations for segmentation. For example, OVDiff [28] synthesizes class-specific support images and extracts their features to form prototypes that guide open-vocabulary segmentation, while DiffSegmenter [60] mines self- and cross-attention maps from pretrained text-to-image diffusion models to localize arbitrary objects. Building on these insights, iSeg [54] applies an entropy-reduction step to self-attention maps and iteratively fuses them with cross-attention to progressively sharpen segmentation, and Diffseg [57] demonstrates that self-attention features alone can be converted directly into masks. Meanwhile, DiffCut [13] frames unsupervised segmentation as a graph-partitioning problem over diffusion features. Our work not only probes the latent representations learned by diffusion models but also leverages them to jointly enhance both image generation and segmentation.

## 3 Seg4Diff: Segmentation for Diffusion

### 3.1 Motivation and Overview

The remarkable success of diffusion models in text-guided image generation [46, 43] is primarily attributed to their attention mechanisms [46, 20], with recent architectures like multi-modal diffusion transformer (MM-DiT) [15] further enhancing performance through joint attention over image and text modalities. Despite these advancements, the precise interactions within these multi-modal attention mechanisms remain largely underexplored. In this regard, our work aims to deeply investigate these internal interactions to elucidate the principles underlying their success, and based on these insights, we propose a simple yet effective training scheme to transfer the learned representations to the semantic segmentation task.

We begin by briefly reviewing rectified flow framework [36] and multi-modal attention mechanism in MM-DiT in Sec. 3.2. Then, in Sec. 3.3, we delve into how text and image tokens interact in attention mechanism of MM-DiT. Building on this, we identify key layers critical for image-text alignment and thereby results in faithful image generation, where we introduce zero-shot segmentation framework to leverage this semantic grounding capability. Subsequently, we propose learning strategy that jointly enhances both segmentation and generative capabilities in Sec. 3.5.

## 3.2 Preliminaries

**Rectified flow framework.** While diffusion models reverse a predefined process using simulated data [22, 52], conditional flow matching (CFM) [35] trains continuous normalizing flows without simulation by framing generative modeling as regression between analytic conditional vector fields and a learnable velocity field. Rectified Flow [36] further simplifies this framework by adopting a deterministic linear interpolation between data and noise distributions. Specifically, given a data sample $x_0 \sim p_{\text{data}}$ and a noise sample $\epsilon \sim \mathcal{N}(0, I)$, the interpolation at time $t \in [0, 1]$ is defined as:

$$x_t = (1 - t)x_0 + t\epsilon, \quad \epsilon \sim \mathcal{N}(0, I). \tag{1}$$

The corresponding ground-truth velocity field along this path is constant and given by

$$u_t(x_t) = \epsilon - x_0.$$

Let $v_t(x_t)$ denote the predicted velocity field parameterized by the model. The training objective minimizes the discrepancy between $v_t(x_t)$ and $u_t(x_t)$, leading to the flow matching (FM) loss

$$\mathcal{L}_{\text{FM}} = \mathbb{E}_{t \sim \mathcal{U}(0,1),\, x_0 \sim p_{\text{data}},\, \epsilon \sim \mathcal{N}(0,I)} \big\| u_t(x_t) - v_t(x_t) \big\|^2. \tag{2}$$

**Multi-modal attention.** Building upon this formulation, transformer-based architecture known as multi-modal diffusion transformer (MM-DiT), which is exemplified by Stable Diffusion 3 (SD3) [15], results in significant improvements in generation capability. At its core, MM-DiT employs a multi-modal attention (MM-Attn) mechanism that simultaneously processes image and text modalities.

Given input image $I$ and text $T$, let $x_{\text{img}} \in \mathbb{R}^{hw \times d}$, and $x_{\text{text}} \in \mathbb{R}^{l \times d}$ be the image and text embeddings, where $h$ and $w$ is the height and width of image latent, and $l$ is the text token length. For each modality $m \in \{I, T\}$, query, key, and value embeddings $Q_m$, $K_m$, and $V_m$ are channel-wise split to $Q_m^h$, $K_m^h$, and $V_m^h$ for each attention head $h \in \{1, 2, \cdots, H\}$, where $H$ is the number of heads. These embeddings are concatenated into $Q^h$, $K^h$ and $V^h \in \mathbb{R}^{(hw+l) \times (d_k)}$ respectively:

$$Q^h = [\,Q_I^h; Q_T^h\,], \quad K^h = [\,K_I^h; K_T^h\,], \quad V^h = [\,V_I^h; V_T^h\,], \tag{3}$$

where $d_k$ is the attention embedding dimension and $[\cdot; \cdot]$ denotes token-wise concatenation. Subsequently, these results are leveraged to produce multi-head attention score $A^h$, which is then multiplied with value tokens to produce attention output $O^h$. The final MM-Attn$(x_{\text{img}}, x_{\text{text}})$ after output projection $\mathcal{P}_O$ is then split back into image and text embeddings for downstream processing:

$$A^h = \text{softmax}\big(Q^h (K^h)^\top / \sqrt{d_k}\big), \tag{4}$$

$$O^h = A^h V^h, \tag{5}$$

$$\text{MM-Attn}(x_{\text{img}}, x_{\text{text}}) = \mathcal{P}_O([A^1 V^1, \ldots, A^H V^H]), \tag{6}$$

where $[\cdot, \cdot]$ denotes channel-wise concatenation. The MM-Attn module is designed to handle four distinct types of query-to-key interaction: image-to-image (I2I), image-to-text (I2T), text-to-image (T2I), and text-to-text (T2T), which we denote $A_{I2I}$, $A_{I2T}$, $A_{T2I}$ and $A_{T2T}$ respectively. These four pathways are illustrated in Fig. 2(a), where we particularly focus on $A_{I2T}$ for analysis.

## 3.3 Emergent Semantic Alignment

In this section, we investigate the interactions between $x_{\text{img}}$ and $x_{\text{text}}$ in DiT. To do so, we decompose and analyze attention scores and the resulting features across all heads and layers. For all the analysis, we sample 50 text prompts consisting of diverse linguistic and stylistic nature from DrawBench [49] and then generate corresponding images using the pretrained model. For clarity, we report results at timestep $t = 8$ and $t = 28$; additional analyses over other timesteps are provided in Appx. B.1. In addition, we focus on Stable Diffusion 3 architecture in our main analysis, where analysis on other MM-DiT variants such as Stable Diffusion 3.5 [53] and Flux [3] can be found in Appx. C.

**Decomposing joint image-text attention.** We start by visualizing and quantifying the attention scores to assess interaction strengths between image and text tokens in MM-Attn. As shown in Fig. 2 (b) and (c), we compute layer-wise attention scores and aggregate them by interaction type, equivalent to Eq. 4. Fig. 2 (b) reveals that I2T scores are disproportionately higher than I2I, even though the I2T region in (a) is roughly 40× smaller than I2I, indicating that I2T dominates the overall attention budget. Conversely, Fig. 2 (c) shows that T2I interactions are relatively smaller than T2T, suggesting that text tokens primarily self-attend to preserve semantic anchors. Based on these findings, we focus our subsequent analysis on I2T interactions, where further discussion can be found in Appx. B.2.

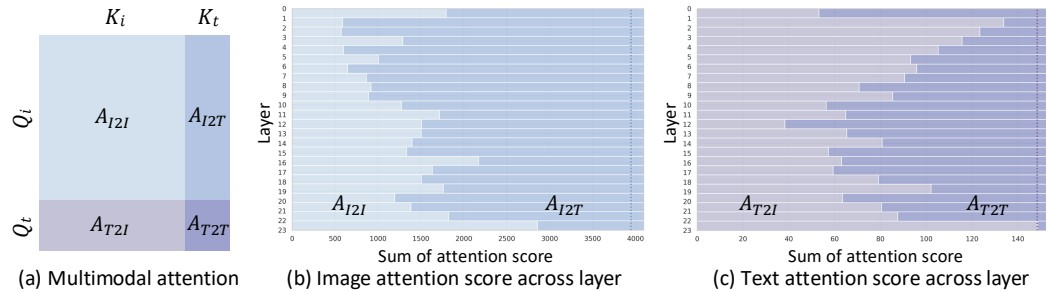

(a) Multimodal attention     (b) Image attention score across layer     (c) Text attention score across layer

Figure 2: **Multi-modal attention mechanism.** (a) Conceptual visualization of the attention map. (b–c) Ratios of attention assigned to image vs. text tokens. The dotted line denotes the ratio under uniform attention. Higher cross-modal proportions are observed in $A_{I2T}$ and $A_{T2I}$.

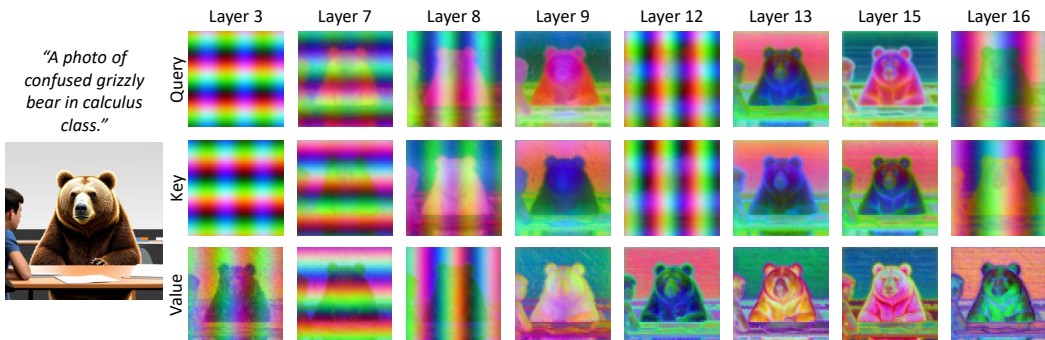

Figure 3: **PCA visualization of query, key and value projections.** PCA results demonstrate that some layers exhibit strong positional bias, whereas some layers show clear semantic groups.

**Attention feature analysis.** To understand *how* MM-DiT's strong image-text interaction operates, we analyze feature space of MM-Attn. Feature PCA visualized in Fig. 3 highlights several layers indicate that query and key features are semantically well aligned, whereas other layers display stronger positional biases. We further analyze the average norm of the value-projected features at each layer, which is computed as $\frac{1}{N}\sum_{i=1}^{N}\|V^i\|_2$ for image, text and actual prompt tokens, which excludes following padding tokens. The attention norm serves as a proxy for the dominance of aggregated information, i.e., how strongly each key token influences a query token [32]. As shown in Fig. 4, similar layers—particularly the 9th MM-DiT block—exhibit notably high attention norms for

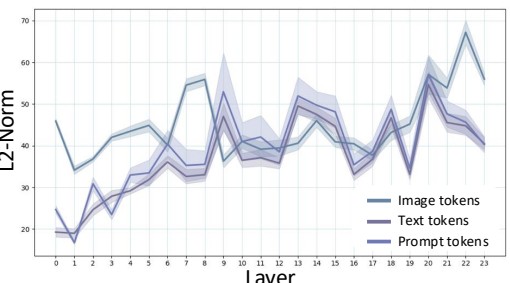

Figure 4: **Attention feature analysis.** The L2 norm of the value projection for image and text tokens reveals that certain layers exhibit significantly stronger value magnitudes for text tokens compared to image tokens.

text tokens compared to that of image tokens. This pattern suggests that text information is injected primarily into semantically aligned image-token regions at these specific layers.

**Attention perturbation for image-text alignment.** Building on this observation, we test whether these layers *causally* drive image synthesis and image–text alignment. Following SEG [25], we apply Gaussian blur to the image-to-text (I2T) attention maps of selected layers and regenerate images, which is shown in Fig. 5. If these layers mediate alignment, attenuating their attention should reduce semantic fidelity. Indeed, perturbing specific layers–which matches with semantically well-aligned layers in our previous analysis–induces clear mismatches between text and content, confirming their critical role in aligning image and text.

We then convert this perturbation into a lightweight guidance mechanism to both validate and exploit the effect. Rather than directly perturbing the sample, we *guide* the standard denoising trajectory

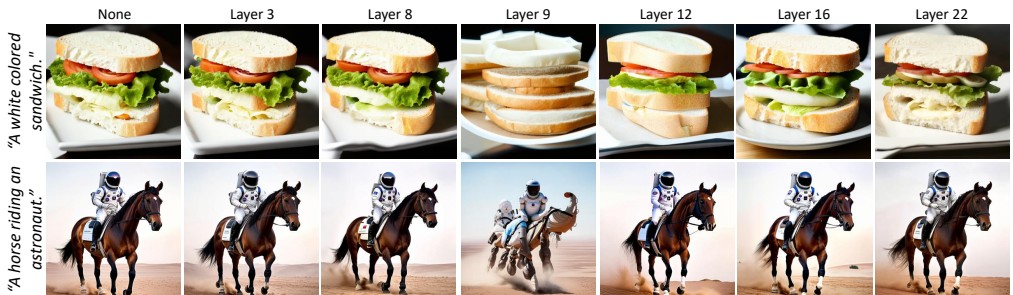

Figure 5: **Effect of layer-wise I2T attention perturbation.** Perturbing specific layer causes pronounced structural degradation, whereas other layers yields minor changes.

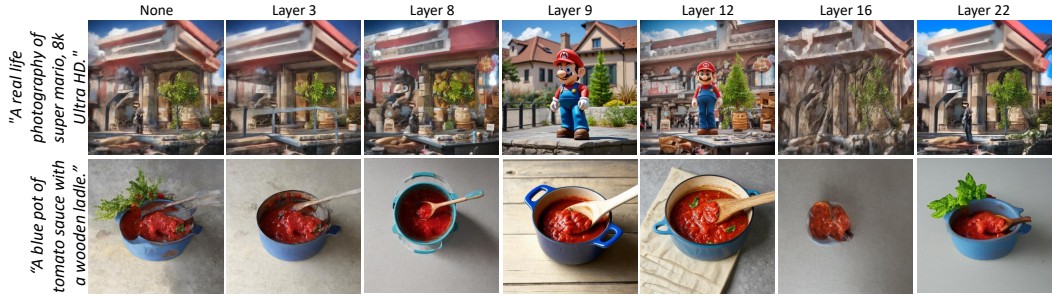

Figure 6: **Effect of perturbed I2T guidance.** Guiding specific layer with perturbed I2T attention score significantly enhances image quality.

with a negatively perturbed companion sample [1, 25]. Specifically,

$$\tilde{\epsilon}_\theta(x_t) = \epsilon_\theta(x_t; A_{I2T}) + s\big(\epsilon_\theta(x_t; A_{I2T}) - \hat{\epsilon}_\theta(x_t; \tilde{A}_{I2T})\big), \tag{7}$$

where $\tilde{\epsilon}_\theta$ is the guided noise estimate, $\epsilon_\theta$ the standard estimate, $\hat{\epsilon}_\theta$ is the perturbed noise estimate with $\tilde{A}_{I2T}$, which is the perturbed version of $A_{I2T}$, and $s > 0$ a guidance scale. As shown in Fig. 6, this procedure substantially improves image quality while modifying only the specific layer's I2T attention regions.

Together, these findings indicate that (i) semantically well-aligned layers are essential for injecting text conditioning into the image, and (ii) overall synthesis quality hinges disproportionately on these layers—consistent with [2], which likewise identifies particular MM-DiT layers as crucial for effective image editing.

## 3.4 Emergent Semantic Grouping

From the previous analysis, we identified MM-DiT layers that exhibit stronger, more critical multimodal interactions than others and encode rich semantic structure. We now examine whether this semantic grounding manifests from a segmentation perspective.

Specifically, we focus on the semantic grounding capability within the I2T attention maps. Obtaining segmentation prediction begins by encoding the input image $I$ into a latent representation $x_{\text{img}}$ using a VAE encoder $\mathcal{E}_{\text{VAE}}$. To ensure reliable segmentation fidelity, we use an intermediate noisy latent, generated by linearly interpolating the clean latent $\mathcal{E}_{\text{VAE}}(I)$ with random noise $\epsilon \sim \mathcal{N}(0, 1)$ at a denoising timestep $t$, following the rectified flow formulation [15, 36]. This technique preserves spatial structure while retaining semantic content, where the ablation on timestep choice can be found in Appx. B.1. Meanwhile, a text prompt $T$ is formed by concatenating the ground-truth classnames existing in the input image (e.g., *"car mountain sky water"*) and encoded into text embeddings $x_{\text{text}}$ via a text encoder $\mathcal{E}_T$:

$$x_{\text{img}} = t \cdot \mathcal{E}_{\text{VAE}}(I) + (1 - t) \cdot \epsilon, \tag{8}$$

$$x_{\text{text}} = \mathcal{E}_T(T). \tag{9}$$

Both embeddings, $x_{\text{img}}$ and $x_{\text{text}}$, are fed into the MM-DiT blocks. In each layer, let $A^h_{I2T}$ be the attention map from image tokens to a specific text token for head $h$. We compute the mean I2T

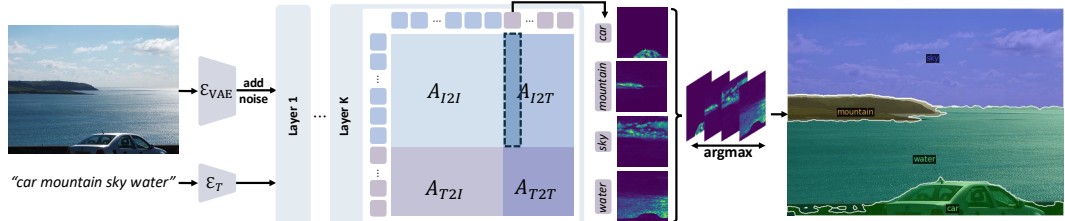

Figure 7: **Open-vocabulary semantic segmentation scheme in our framework.** We generate segmentation masks by interpreting the I2T attention scores, where the score map for each text token serves as a direct measure of image-text similarity to produce the final prediction.

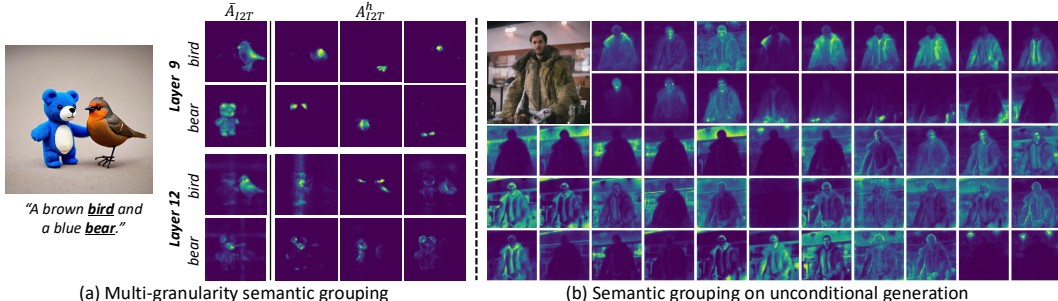

(a) Multi-granularity semantic grouping    (b) Semantic grouping on unconditional generation

Figure 9: **Deeper analysis on multi-modal attention mechanism.** (a) multi-granularity behavior of token-level and head-level attention, and (b) emergent semantic grouping on <pad> tokens in unconditional generation scenario.

attention map, $\bar{A}_{I2T} \in \mathbb{R}^{hw \times l \times Hd_k}$, by averaging I2T attention $A_{I2T}^h \in \mathbb{R}^{hw \times l \times d_k}$ across all heads, which is then reshaped to construct a mask logit $M^{(j)} \in \mathbb{R}^{h \times w \times l}$ corresponding to each text token index $j$. Formally, the overall process is summarized by the following equations:

$$\bar{A}_{I2T} = \frac{1}{H} \sum_{k=1}^{H} A_{I2T}^h, \tag{10}$$

$$M^{(j)} = \text{Reshape}(\bar{A}_{I2T}^{(j)}) \in \mathbb{R}^{h \times w}, \quad \text{for } j = 1, \dots, l. \tag{11}$$

Since a single classname may correspond to multiple text tokens, we average their respective attention maps to produce a single logit map per class. This results in a final logit tensor $P \in \mathbb{R}^{h \times w \times C}$, where $C$ is the number of entire classes. The final prediction is obtained by applying an argmax operation at each pixel location to assign the most likely class. [71, 11, 51]

We visualize the qualitative results in Fig. 7 and quantitative results in Fig. 8. Detailed explanation regarding evaluation metrics, pACC, mACC and mIoU, can be found in Appx. A.4. The result shows that specific layers are responsible for semantic grounding, where 9th layer performs the best, which aligns with previously identified in above. We refer to layers as *semantic grounding expert* layers.

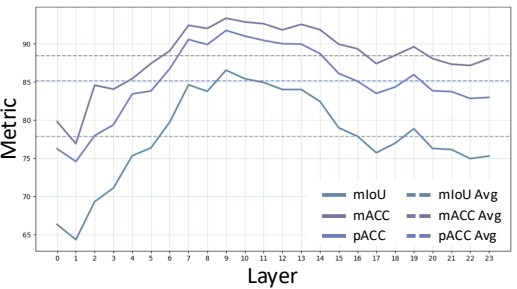

Figure 8: **Segmentation performance across layers.** Semantic grounding quality varies across MM-DiT layers, peaking in the middle blocks and specifically at the 9th layer.

**Multi-granularity semantic grouping in multi-head attention.**    We further decompose attention map of semantic grounding expert layer into individual heads to see how each text token captures complete semantic region. Fig. 9 (a) visualizes head- and token-level attention map of for the highlighted text token, $A_{I2T}^h$ and $A_{I2T}$ respectively. The attention heads focus on a distinct semantic part of the object, such as ears and legs of a bear. Subsequently, token-level attention map precisely delineates the target semantics, demonstrating an inherent multi-granular grouping capability.

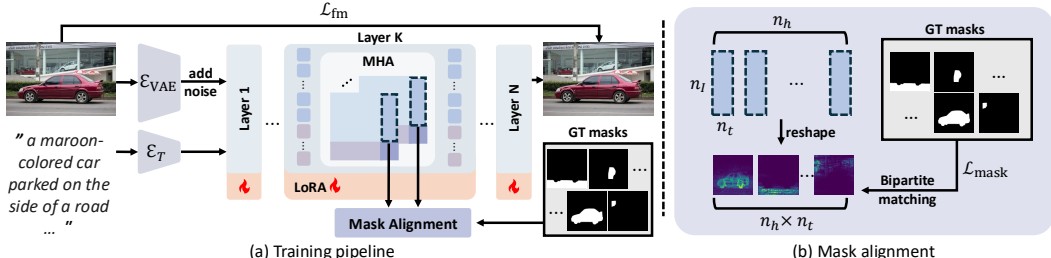

(a) Training pipeline               (b) Mask alignment

Figure 10: **Lightweight fine-tuning pipeline via mask alignment.** We introduce a simple yet effective *mask alignment* strategy that strengthens the I2T attention maps in the semantic grounding expert layer during additional diffusion fine-tuning with a LoRA adapter.

**Unsupervised semantic grouping in `<pad>` tokens.** This grouping behavior also emerges during unconditional generation, where the entire text sequence is set to `<pad>`. As shown in Fig. 9 (b), attention maps from individual `<pad>` tokens—despite lacking explicit semantics—consistently attend to coherent regions, with different token positions specializing in different objects or parts. This allows the model to discover and segment meaningful groups, and these attention maps can thus serve as effective proposals for training-free unsupervised segmentation. To leverage this, we adapt the segmentation scheme in Fig. 7. Instead of using the attention map of a classname token, we select the best-matching mask from the proposals generated by the `<pad>` tokens.

### 3.5 Boosting Segmentation and Generation

Motivated by our layer-wise analysis and the causal perturbation study in Sec. 3.3, which identified a specific semantic grounding expert layer in MM-DiT as the key mediator of image–text alignment, we introduce a simple, lightweight LoRA fine-tuning scheme, called *mask alignment for segmentation and generation (MAGNET)*, that directly strengthens alignment in this layer. The goal is to boost semantic grouping in its I2T attention while preserving the model's generation quality; empirically, this improves both zero-shot segmentation and image synthesis.

Specifically, we optimize two complementary losses. First, for each prediction $v_t(x_t)$ we apply a flow-matching loss $\mathcal{L}_{\text{FM}}$ to supervise the diffusion process. Second, to enhance the semantic grouping expressed by the expert layer's I2T attention, we add a mask loss $\mathcal{L}_{\text{mask}}$.

Specifically, to compute $\mathcal{L}_{\text{mask}}$, we extract the $l \cdot H$ I2T attention maps from the semantic grounding expert layer for $l$ text tokens and $H$ heads and normalize each map to form candidate mask logits. We then perform bipartite matching between these candidates and the ground-truth masks to obtain one-to-one assignments [8]. Since we do not consider classification, we match masks solely based on the mask loss. For each matched pair, the loss is computed similarly as bipartite matching criterion:

$$\mathcal{L}_{\text{mask}} = \lambda_{\text{focal}} \mathcal{L}_{\text{focal}} + \lambda_{\text{dice}} \mathcal{L}_{\text{dice}}, \tag{12}$$

where $\mathcal{L}_{\text{focal}}$ is focal loss, $\mathcal{L}_{\text{dice}}$ is Dice loss, and $\lambda_{\text{focal}}, \lambda_{\text{dice}}$ weight their contributions.

Finally, the total objective is

$$\mathcal{L}_{\text{total}} = \mathcal{L}_{\text{FM}} + \lambda_{\text{mask}} \mathcal{L}_{\text{mask}}, \tag{13}$$

with $\lambda_{\text{mask}}$ controlling the strength of mask alignment.

## 4 Experiments

### 4.1 Implementation Details

For zero-shot inference, the diffusion process is fixed at timestep $t = 8$ of 28 using the flow-matching Euler discrete scheduler. Training uses 10k images from either SA-1B [30] or COCO [34], with captions generated by CogVLM [61] following SD3's procedure [15]. Head-level attention maps are used for SA-1B and token-level maps for COCO to match mask granularity. Images are processed at $1024 \times 1024$ resolution, and each transformer layer is equipped with a LoRA module of rank $r = 16$, trained using AdamW with lr $= 1 \times 10^{-5}$, default $\beta$ parameters, and weight decay. Training runs on two NVIDIA A6000 GPUs with per-device batch size 4 and gradient accumulation for an effective batch size of 16.

| Model | Arch. | Train. | VOC20 | Object | PC59 | ADE | City |
|---|---|---|---|---|---|---|---|
| ProxyCLIP [33] | CLIP-H/14 | – | 83.3 | 49.8 | 39.6 | 24.2 | 42.0 |
| CorrCLIP [66] | CLIP-H/14 | – | 91.8 | 52.7 | 47.9 | 28.8 | 49.9 |
| DiffSegmenter [60] | SD1.5 | – | 66.4 | 40.0 | 45.9 | 24.2 | 12.4 |
| iSeg [54] | SD1.5 | – | 82.9 | 57.3 | 39.2 | 24.2 | 24.8 |
| **Seg4Diff** | SD3 | – | 89.2 | 62.0 | 49.0 | 34.2 | **26.5** |
| **Seg4Diff** | SD3.5 | – | 86.1 | 57.8 | 43.4 | 30.7 | 23.8 |
| **Seg4Diff** | Flux.1-dev | – | 83.1 | 50.6 | 38.2 | 23.9 | 17.1 |
| **Seg4Diff + MAGNET** | SD3 | SA-1B | 89.1 | 62.0 | 49.1 | 34.7 | 25.4 |
| **Seg4Diff + MAGNET** | SD3 | COCO | 89.8 | 62.9 | 51.2 | 35.2 | 26.0 |

Table 1: **Open-vocabulary semantic segmentation performance.** Cross-modal alignment in the I2T attention maps of the semantic grounding expert layer yields competitive results, further enhanced by mask alignment.

| Model | Arch. | Train. | VOC21 | PC59 | Object | Stuff-27 | City | ADE |
|---|---|---|---|---|---|---|---|---|
| ReCO [50] | CLIP-L/14 | - | 25.1 | 19.9 | 15.7 | 26.3 | 19.3 | 11.2 |
| MaskCLIP [14] | CLIP-B/16 | - | 38.8 | 23.6 | 20.6 | 19.6 | 10.0 | 9.8 |
| MaskCut [62] | DINO-B/8 | - | 53.8 | 43.4 | 30.1 | 41.7 | 18.7 | 35.7 |
| DiffSeg [57] | SD1.5 | - | 49.8 | 48.8 | 23.2 | 44.2 | 16.8 | 37.7 |
| DiffCut [13] | SSD-1B [19] | - | **62.0** | **54.1** | 32.0 | 46.1 | **28.4** | 42.4 |
| **Seg4Diff** | SD3 | - | 54.9 | 52.6 | 38.5 | 49.7 | 24.2 | 44.9 |
| **Seg4Diff** | SD3.5 | - | 52.3 | 52.9 | 36.8 | 47.1 | 24.2 | 41.5 |
| **Seg4Diff + MAGNET** | SD3 | SA-1B | 55.1 | 52.8 | **39.0** | 50.8 | 24.2 | 45.0 |
| **Seg4Diff + MAGNET** | SD3 | COCO | 56.1 | 53.5 | 38.8 | 53.5 | 24.4 | 45.4 |

Table 2: **Unsupervised segmentation performance.** Although not specifically designed for unsupervised semantic segmentation, exploiting the emergent semantic grouping of `<pad>` tokens in the I2T attention maps achieves competitive results.

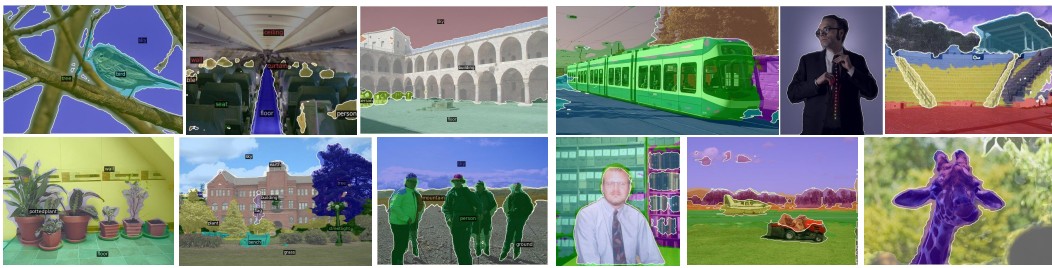

(a) Open-vocabulary semantic segmentation     (b) Unsupervised segmentation

Figure 11: **Qualitative results on segmentation.** Through our Diff4Seg, MM-DiT demonstrates strong performance in (a) open-vocabulary semantic segmentation and (b) unsupervised segmentation. Additional results are provided in Appx. F.1.

## 4.2 Seg4Diff for Segmentation

**Experimental setting.** We evaluate our method on two tasks: open-vocabulary semantic segmentation and unsupervised segmentation. For open-vocabulary semantic segmentation, we report mIoU on the validation sets of PascalVOC [16], COCO-Object [34], Pascal Context-59 [16], and ADE20K [69], excluding the background class. We compare U-Net diffusion-based segmentation methods [60, 55] as well as CLIP-based approaches [33, 66]. We note that CLIP-based methods process entire classnames for prediction, which is not identical to our evaluation setting. For unsupervised segmentation, we report mIoU on the validation sets of PascalVOC, Pascal Context-59, COCO-Object, COCO-Stuff-27 [34], Cityscapes [12], and ADE20K. We adopt mask proposal evaluation protocol of DiffSeg [57], including the background class for PascalVOC, Pascal Context-59, and COCO-Object in line with DiffCut [13]. We compare with training-free methods built on diverse backbones, including CLIP [50, 70], DINO [62], and U-Net diffusion models [57, 13].

**Results.** Tab. 1 summarizes our open-vocabulary semantic segmentation results. Remarkably, using only this single layer without any refinement or postprocessing, our approach achieves competitive performance on Pascal VOC and COCO-Object dataset. Moreover, ours are robust on more complex datasets, where DiT architecture benefits from the larger and consistent spatial resolution throughout the entire layers. This result shows that MM-DiT inherently learned fine-grained semantic grounding capability during the generation process. On the other hand, as shown in Tab. 2, our method achieves competitive performance on unsupervised segmentation. This strong outcome indicates that an emergent knowledge of semantic grouping exists within the model's multi-modal attention layers, which effectively function as learnable proxies for semantic classes—even when occupied by content-free `<pad>` tokens.

## 4.3 Seg4Diff for Boosting Segmentation and Generation

**Experimental setting.** We evaluate the image generation capability of Seg4Diff on three benchmark datasets: Pick-a-pic [31], MS-COCO [34], and SA-1B captions generated via CogVLM [61]. To assess text-image coherence, we utilize CLIPScore [21], which measures the semantic alignment between the text prompt and the generated image. We report T2I-CompBench++ [27] to further evaluate the fidelity and compositional quality of our generated images.

| Method | Training | Pick-a-Pic | COCO | SA-1B | Mean |
|---|---|---|---|---|---|
| Baseline | – | 27.0252 | 26.0638 | 28.3422 | 27.1437 |
| + MAGNET | SA-1B | **27.0547** | 26.2318 | 28.4476 | 27.2447 |
| + MAGNET | COCO | 27.0409 | **26.2319** | **28.5553** | **27.2760** |

Table 3: **CLIPScore on text-to-image generation benchmarks.** Mask alignment consistently improves alignment with text prompts across various datasets.

| Method | Training | Attribute binding | | | Object relationships | | | Num. | Comp. |
|---|---|---|---|---|---|---|---|---|---|
| | | Color | Shape | Texture | 2D | 3D | non | | |
| Baseline | – | 0.7864 | 0.5644 | 0.7200 | **0.2435** | **0.3318** | **0.3124** | 0.5566 | 0.3719 |
| + MAGNET | SA-1B | 0.7836 | 0.5679 | 0.7252 | 0.2330 | 0.3151 | 0.3113 | 0.5460 | 0.3709 |
| + MAGNET | COCO | **0.7919** | **0.5687** | **0.7260** | 0.2301 | 0.3234 | 0.3120 | **0.5584** | **0.3735** |

Table 4: **T2I-Compbench++ performance.** Mask alignment enhances attribute binding, object relationships, and compositional understanding compared to the baseline.

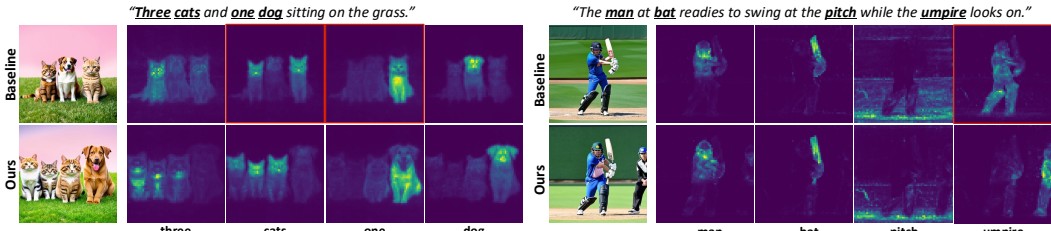

Figure 12: **Effects of the proposed mask alignment.** Mask alignment improves structural coherence and alignment between image and text.

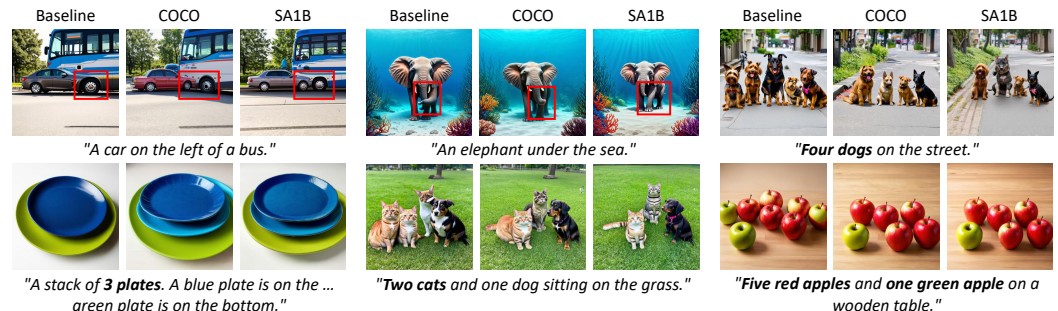

Figure 13: **Qualitative results on image generation.** Mask alignment improves compositional accuracy by reducing structural errors, correcting object counts, and refining colors and textures.

**Results.** As shown in Tab. 3 and 4, our fine-tuning method improves image generation quality over the baseline, achieving higher CLIPScores and better text–image alignment. On T2I-Compbench++, the COCO-trained model shows clear gains in attribute binding, numerical concepts, and complex scenes, which suggests our method effectively enhances the model's ability to render objects with their correct attributes. However, because our training scheme do not target reasoning on object interactions, the baseline model maintains a marginal lead in rendering object relationships. Qualitative results in Fig.12 and 13 further confirm that ABoost corrects attention misalignments of the baseline, producing more accurate and plausible images. Overall, our strategy effectively enhances object-centric image generation as well as segmentation performance, as previously evidenced in Tab. 1 and 2.

## 5 Conclusion

This paper introduces Seg4Diff, a systematic framework that investigates the emergent capabilities of multi-modal diffusion transformers towards open-vocabulary semantic segmentation. Through an in-depth analysis of multi-modal attention mechanism, we identify semantic grounding expert layers that are critical for aligning textual semantics with corresponding image regions. We demonstrate that attention maps from these layers can be directly used to achieve competitive zero-shot segmentation performance. To further enhance this capability, we introduce a lightweight fine-tuning method that strengthens the semantic grouping in these expert layers. This targeted approach not only leads to meaningful improvements in segmentation quality but also enhances the model's generative fidelity. Our findings reveal that semantic alignment is an intrinsic property of diffusion transformers that can be selectively amplified with minimal computational cost. This work paves the way toward unified models that bridge the gap between generation and perception, delivering both high-quality image synthesis and accurate semantic understanding.

## Acknowledgments and Disclosure of Funding

This research was supported by Institute of Information & communications Technology Planning & Evaluation (IITP) grant funded by the Korea government (MSIT) (RS-2019-II190075, RS-2024-00509279, RS-2025-II212068, RS-2023-00227592, RS-202502214479, RS-2024-00457882, RS-2025-25441838, RS-2025-25441838, RS-2025-02214479, RS-2025-02217259), the Culture, Sports, and Tourism R&D Program through the Korea Creative Content Agency grant funded by the Ministry of Culture, Sports and Tourism (RS-2024-00345025, RS-2024-00333068, RS-2023-00222280, RS-2023-00266509), and National Research Foundation of Korea (RS-2024-00346597), and the ETH AI Center through an ETH AI Center postdoctoral fellowship to Sunghwan Hong.

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

# Appendix

# A Further implementation detail

## A.1 Model configuration

We utilize the Stable Diffusion 3 (SD3) [15] model for our main analysis. SD3 originally incorporates three vision-language text encoders: CLIP-G/14, CLIP-L/14 [44], and T5-XXL [45]. Due to memory constraints, we disable the T5 encoder and use only the first 77 tokens from the two CLIP encoders. We leverage the attention score from timestep $t = 8$ of 28, where the attention logits are semantically grouped well. Further ablation on timesteps can be found at Sec. B.1. We used classifier-free guidance [23] with scale 7.5 for generation if not specified.

For training and segmentation evaluation, the input images are center-cropped if non-square and resized to a resolution of $1024 \times 1024$. After VAE encoding, the image is further downsampled to $64 \times 64$ latent. The resulting attention maps are bilinear upsampled back to the original image size for segmentation evaluation.

## A.2 Text prompts for analysis

We borrowed text prompts from DrawBench [49], where diverse categories are included to assess the capability of generative models. From 200 prompts in total, we randomly sampled 50 prompts for our analysis. The full list of selected prompts is shown in Fig.14.

```
1.  A red colored car.
2.  A black colored dog.
3.  A blue colored dog.
4.  A red colored banana.
5.  A white colored sandwich.
6.  A yellow colored giraffe.
7.  A green cup and a blue cell phone.
8.  A horse riding an astronaut.
9.  A shark in the desert.
10. Three cars on the street.
11. One dog on the street.
12. Two dogs on the street.
13. One cat and one dog sitting on the grass.
14. Three cats and one dog sitting on the grass.
15. Three cats and two dogs sitting on the grass.
16. A triangular pink stop sign.  A pink stop sign in the
shape of a triangle.
17. An illustration of a small green elephant standing
behind a large red mouse.
18. A small blue book sitting on a large red book.
19. A stack of 3 cubes.  A red cube is on the top, sitting
on a red cube.  The red cube is in the middle, sitting on a
green cube.  The green cube is on the bottom.
20. A stack of 3 books.  A green book is on the top,
sitting on a red book.  The red book is in the middle,
sitting on a blue book.  The blue book is on the bottom.
21. A small vessel propelled on water by oars, sails, or an
engine.
22. A large plant-eating domesticated mammal with solid
hoofs and a flowing mane and tail, used for riding, racing,
and to carry and pull loads.
23. An American multinational technology company that
focuses on artificial intelligence, search engine, online
advertising, cloud computing, computer software, quantum
computing, e-commerce, and consumer electronics.
24. A large thick-skinned semiaquatic African mammal, with
massive jaws and large tusks.
25. A machine resembling a human being and able to
replicate certain human movements and functions automatically.
26. A grocery store refrigerator has pint cartons of milk
on the top shelf, quart cartons on the middle shelf, and
gallon plastic jugs on the bottom shelf.

27. An elephant is behind a tree.  You can see the trunk on
one side and the back legs on the other.
28. A pear cut into seven pieces arranged in a ring.
29. Rbefraigerator.
30. Dininrg tablez.
31. An instqrumemnt used for cutting cloth, paper, axdz
othr thdin mteroial, consamistng of two blades lad one on
tvopb of the other and fhastned in tle mixdqdjle so as to
bllow them txo be pened and closed by thumb and fitngesr
inserted tgrough rings on kthe end oc thei vatndlzes.
32. A bicycle on top of a boat.
33. A car on the left of a bus.
34. Acersecomicke.
35. Artophagous.
36. Backlotter.
37. A photo of a confused grizzly bear in calculus class.
38. Photo of an athlete cat explaining it's latest scandal
at a press conference to journalists.
39. Hyper-realistic photo of an abandoned industrial site
during a storm.
40. A real life photography of super mario, 8k Ultra HD.
41. Colouring page of large cats climbing the eifel tower
in a cyberpunk future.
42. Photo of a mega Lego space station inside a kid's
bedroom.
43. A spider with a moustache bidding an equally gentlemanly
grasshopper a good day during his walk to work.
44. A bridge connecting Europe and North America on the
Atlantic Ocean, bird's eye view.
45. A magnifying glass over a page of a 1950s batman comic.
46. A realistic photo of a Pomeranian dressed up like a
1980s professional wrestler with neon green and neon orange
face paint and bright green wrestling tights with bright
orange boots.
47. A sign that says 'Hello World'.
48. A sign that says 'Diffusion'.
49. New York Skyline with 'NeurIPS' written with fireworks
on the sky.
50. New York Skyline with 'Google Research Pizza Cafe'
written with fireworks on the sky.
```

Figure 14: **Selected prompts for our analysis.**

## A.3 Attention perturbation

To assess the importance of image-to-text (I2T) attention alignment across layers, we applied a Gaussian blur along the text token dimension of the I2T attention map. We used 1D Gaussian kernel with standard deviation $\sigma = 9$ and kernel size $k = 5$ to accommodate the typical text token length. The blur was applied to the attention logits after softmax, using reflective padding to preserve the total attention mass per image token.

### A.4 Segmentation evaluation metrics

To assess layer-wise segmentation performance in Sec. 3.4, we use three standard metrics: pixel accuracy (pACC), mean accuracy (mACC), and mean Intersection-over-Union (mIoU). Each provides a progressively more rigorous evaluation based on pixel-level true positives (TP), false positives (FP), and false negatives (FN) for the number classes $N$. pACC reports the fraction of correctly classified pixels over entire classes, which can be easily skewed by large classes like background. mACC averages per-class accuracy, $\frac{TP}{TP+FN}$, treating all classes equally yet still ignoring FP; mIoU is the most comprehensive, computing intersection-over-union, $\frac{TP}{TP+FP+FN}$.

$$\text{pACC} = \frac{\sum TP}{\sum TP + FN} \qquad \text{mACC} = \frac{1}{N} \sum \frac{TP}{TP + FN} \qquad \text{mIoU} = \frac{1}{N} \sum \frac{TP}{TP + FP + FN} \quad (14)$$

For unsupervised segmentation, we forward a noised input image with a null prompt to obtain the <pad> token attention maps. These maps are then greedily merged based on KL-divergence, following the procedure in [57]. The resulting mask proposals are evaluated via bipartite matching with ground-truth masks, while unmatched proposals are treated as false negatives.

### A.5 Further details on image quality metrics.

For Pick-a-Pic, we generate 500 images for five random seeds respectively. We generated 5,000 images for MS-COCO and 1,000 images for SA-1B captions.

## B Additional experiments

### B.1 Ablation on timestep choice

Fig. 15 shows segmentation performance throughout different timesteps applied to the input image. Both PascalVOC and COCO-Object demonstrates the best performance on $t = 8$ of 28, where we report the segmentation performance.

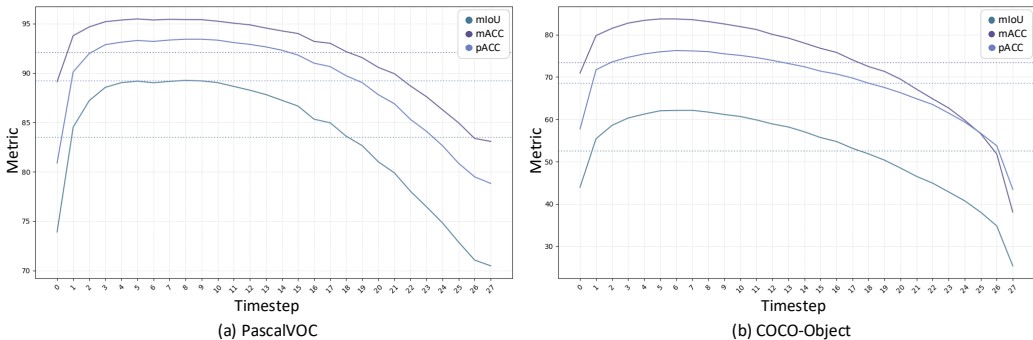

(a) PascalVOC      (b) COCO-Object

Figure 15: **Segmentation performance across denoising timesteps.**

### B.2 Comparison on I2T and T2I attention maps

Fig. 16 presents attention maps for I2T and T2I directions corresponding to the highlighted keywords. The I2T maps exhibit more complete and contiguous object masks, whereas T2I tends to capture only partial or attenuated regions. This discrepancy likely stems from the distinct aggregation roles: I2T attention directly updates image tokens based on textual queries, while T2I updates text tokens conditioned on visual features. Given that the diffusion process ultimately operates on image tokens to synthesize outputs, it is reasonable that I2T attention aligns more strongly with semantically grounded image regions. Although this observation does not constitute a formal proof, the consistency of the qualitative patterns, which is further visualized in Appx. D.1 across prompts supports this interpretation.

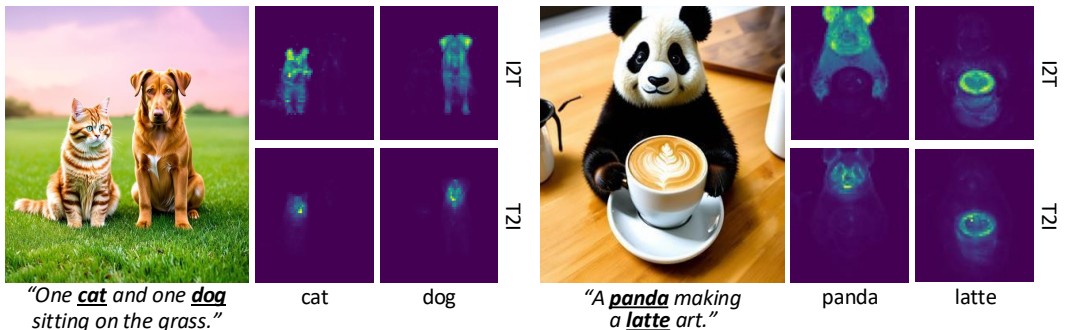

"One **_cat_** and one **_dog_** sitting on the grass."    cat    dog      "A **_panda_** making a **_latte_** art."    panda    latte

Figure 16: **Comparison on I2T and T2I attention maps.**

## B.3 Ablation on normalization method

We conduct an ablation study to determine the optimal method for normalizing attention scores. As shown in Table 5, we evaluate the open-vocabulary semantic segmentation performance across four configurations: using scores before or after the softmax function, with and without additional min-max normalization to scale the logit between 0 to 1. The results indicate that using the raw scores directly after the softmax function yields the best performance. Consequently, we adopt this scheme for all subsequent experiments.

| softmax | min-max | VOC | Object |
|:---:|:---:|:---:|:---:|
| ✗ | ✗ | 83.2 | 55.1 |
| ✗ | ✓ | 85.2 | 53.5 |
| ✓ | ✗ | **89.0** | **61.8** |
| ✓ | ✓ | 88.3 | 57.0 |

Table 5: **Ablation on attention score normalization methods.**

## B.4 Segmentation performance per attention layer and head

We present semantic segmentation results obtained by leveraging the attention scores from individual layers and heads in Fig.17. For a comprehensive evaluation, we measured mIoU on the PASCAL VOC dataset [16], including the background class. Our analysis reveals that the middle layers consistently exhibit superior segmentation performance compared to early or late layers.

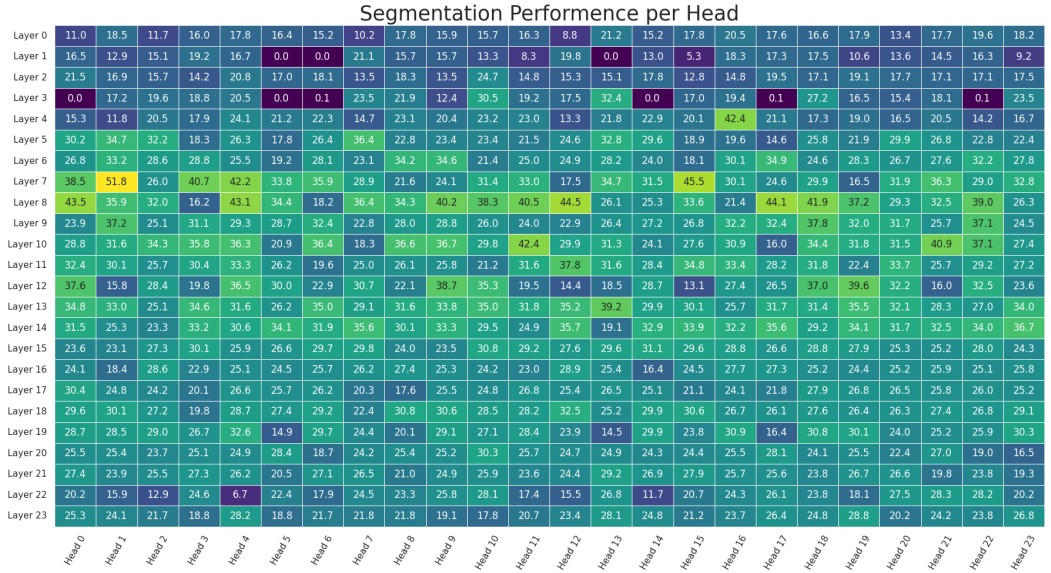

Figure 17: **mIoU score of each head.**

## B.5 Quantitative results on attention perturbation

We use CLIP-I, CLIP-T, and DINO scores to evaluate generation quality and alignment. CLIP-I measures cosine similarity between CLIP embeddings of generated and reference images, reflecting high-level perceptual fidelity. CLIP-T compares the CLIP embedding of a generated image with that of the conditioning text, assessing image–text alignment. DINO instead computes cosine similarity between self-supervised vision embeddings of generated and reference images, offering a language-free measure of semantic similarity. Higher values indicate better fidelity or alignment, with CLIP-I and DINO focusing on image–image consistency and CLIP-T on image–text consistency.

Aligned with the qualitative analysis in Sec. 3.3, 9th layer, which we designated as a semantic grounding expert in SD3, shows a noticeable drop on image fidelity score when perturbed.

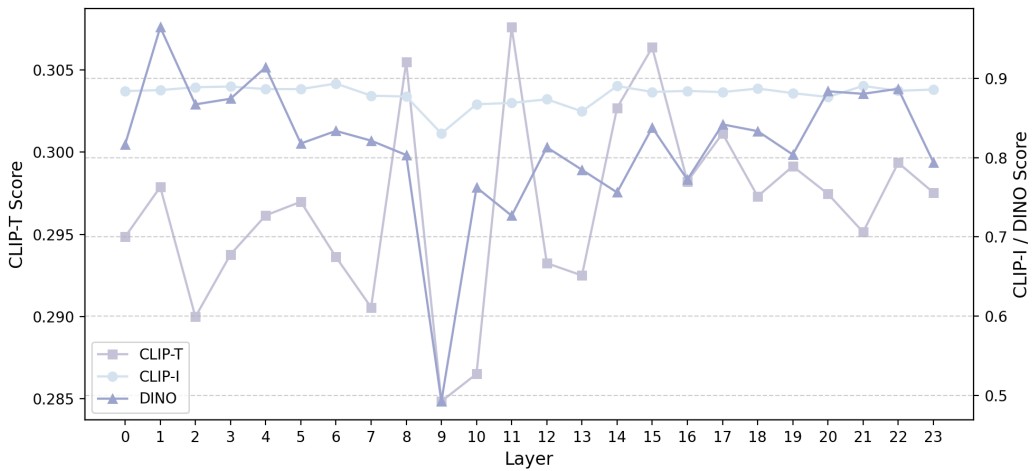

Figure 18: **Image fidelity scores under layer-wise perturbations on SD3.**

## C  Generalization to other baselines

While we mainly leverage Stable Diffusion 3 (SD3) [15] in our main analysis, we also apply our analysis to the other MM-DiT variants, Stable Diffusion 3.5 (SD3.5) [53] and Flux.1-dev [3]. We can similarly observe the correlation between value norm and segmentation performance among layers for both models. While SD3.5 appears to have layer 9, identical to SD3, to exhibit strong semantic grounding, Flux shows layer 12 and 17 to have a similar tendency. This hints that our observation and methodology are not proprietary for SD3, but can be applied to other DiT-based diffusion models with multi-modal attention, highlighting the generalizability of our insights and findings.

## C.1 Stable Diffusion 3.5

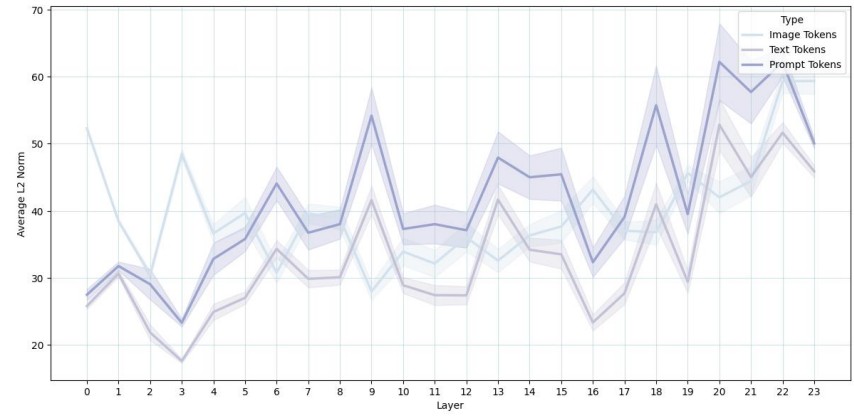

Figure 19: **Average L2-Norm of values across layers on SD3.5.**

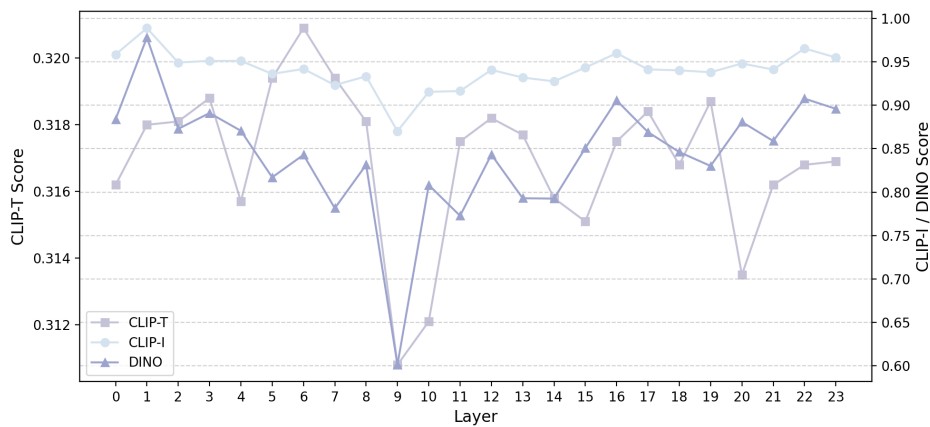

Figure 20: **Image fidelity scores under layer-wise perturbations on SD3.5.**

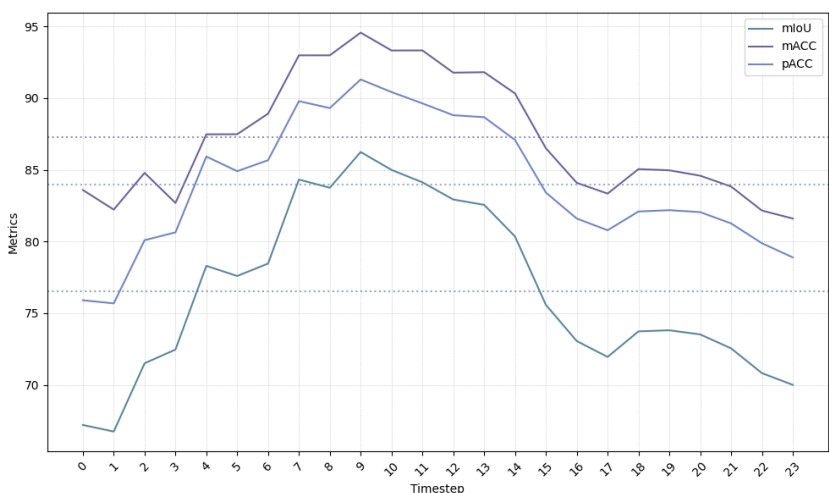

Figure 21: **Segmentation performance across layers on SD3.5.**

## C.2 Flux.1-dev

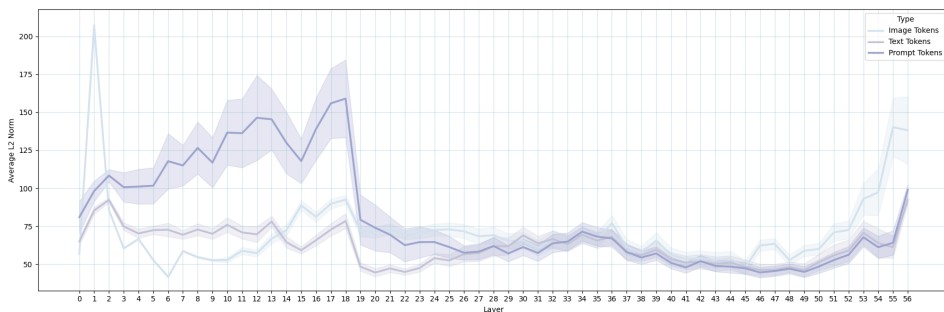

Figure 22: **Average L2-Norm of values across layers on Flux.**

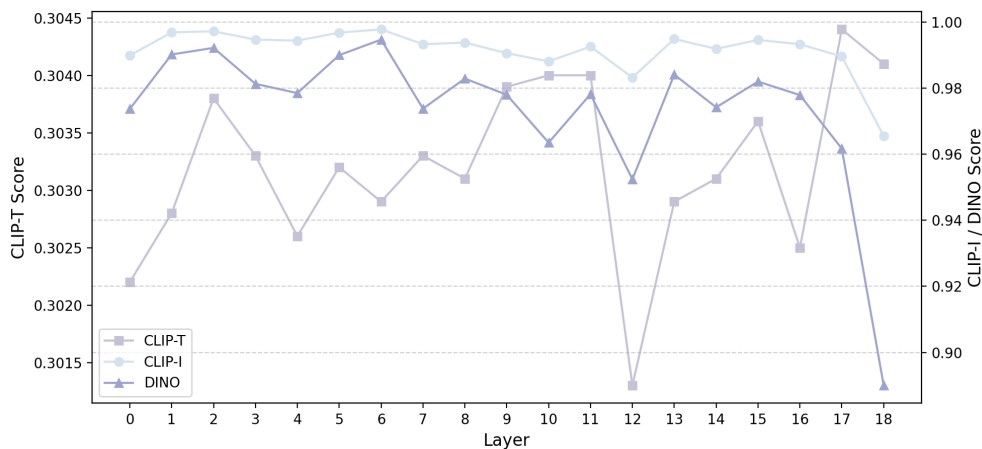

Figure 23: **Image fidelity scores under layer-wise perturbations on Flux.** We evaluate only for the first 19 layers, which employs MM-DiT architecture.

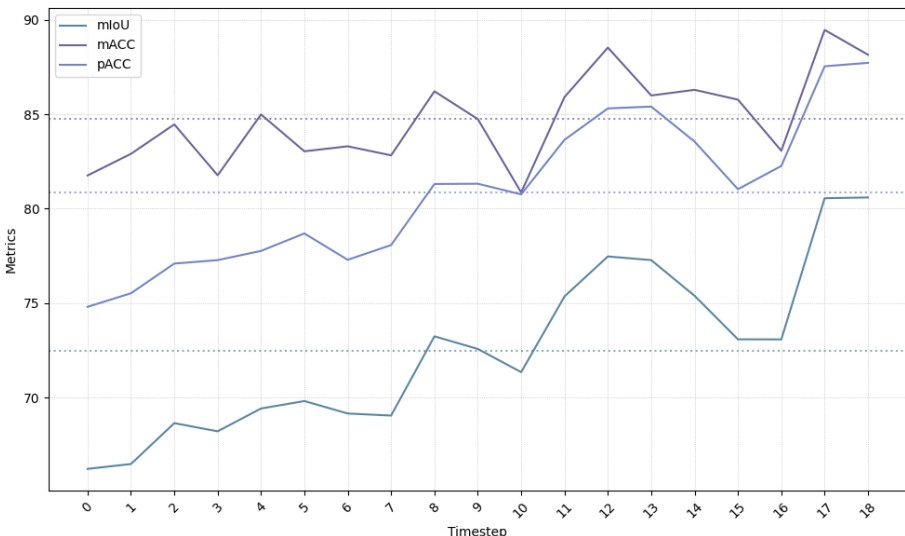

Figure 24: **Segmentation performance across layers on Flux.**

# D  Additional visualizations

## D.1  Image-to-text (I2T) and text-to-image (T2I) attention map

We provide extended visualizations of image-to-text (I2T) and text-to-image (T2I) attention maps in Fig. 25. The maps are taken from layer 9, where we observe strong semantic alignment between visual and textual modalities. These results offer insight into the emergent cross-modal grounding dynamics of the multi-modal diffusion transformer (MM-DiT).

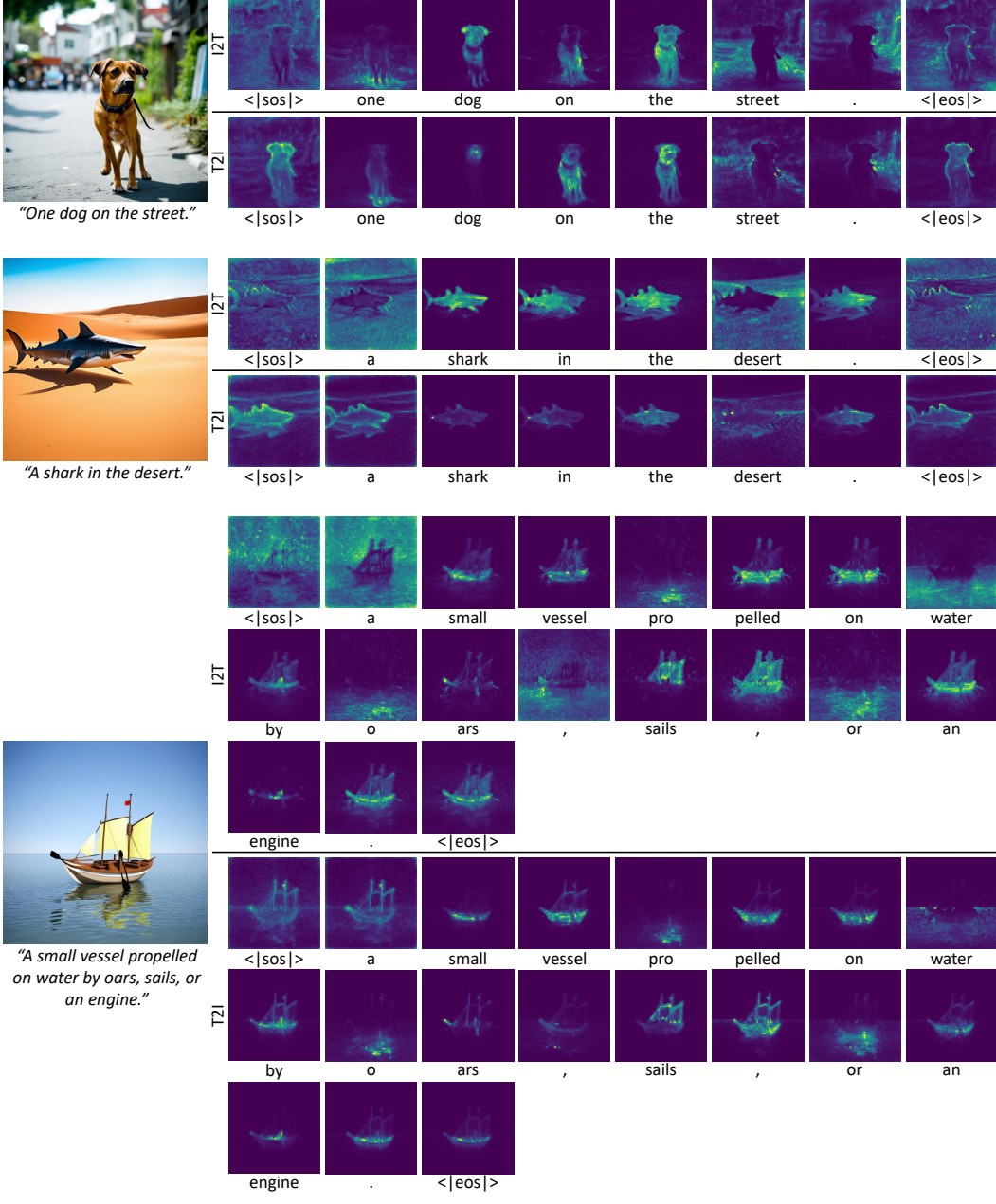

Figure 25: **Image-to-Text attention map visualization of all prompt tokens.**

## D.2 Per-head attention maps

We present comprehensive visualizations of attention maps for individual heads in selected layers, as shown in Fig.26 and Fig.27. These results illustrate that each attention head focuses on distinct image regions. This effect is particularly pronounced in layer 9, where individual heads attend to different parts of the corresponding semantic region.

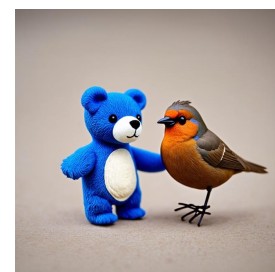

*"A brown bird and a blue **bear**."*

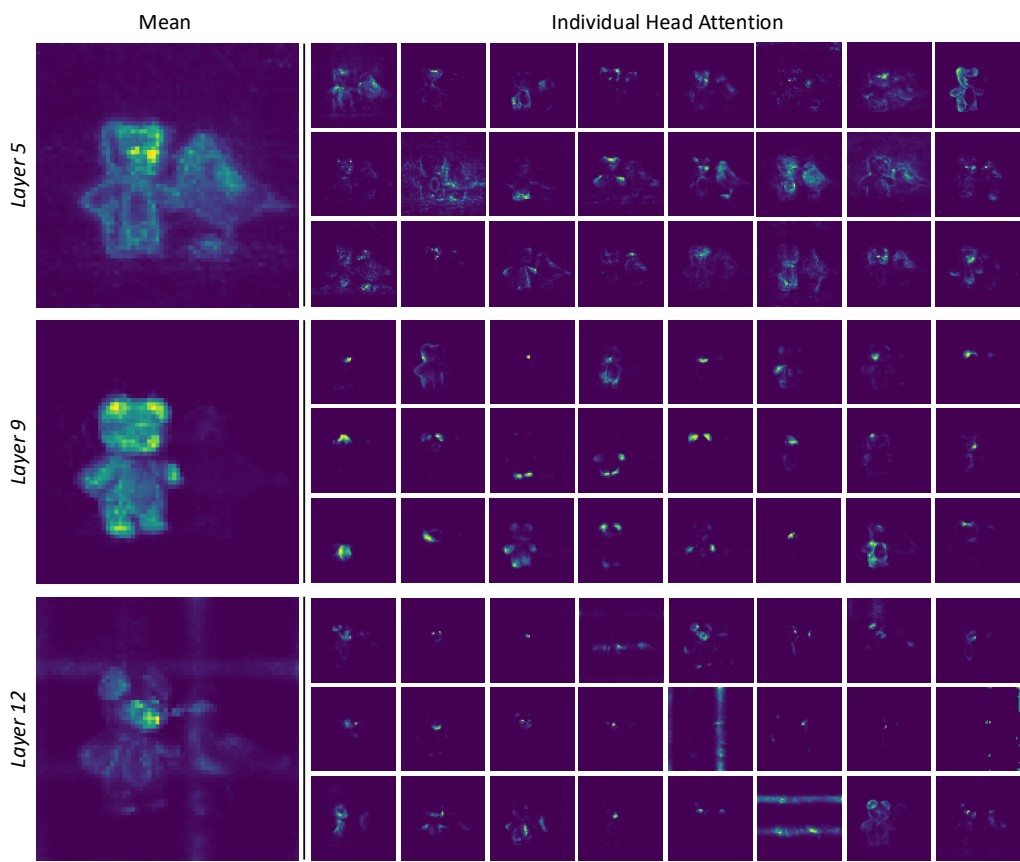

Figure 26: **Visualization of attention scores for all heads.**

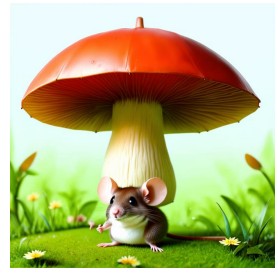

*"Illustration of a mouse using a __mushroom__ as an umbrella."*

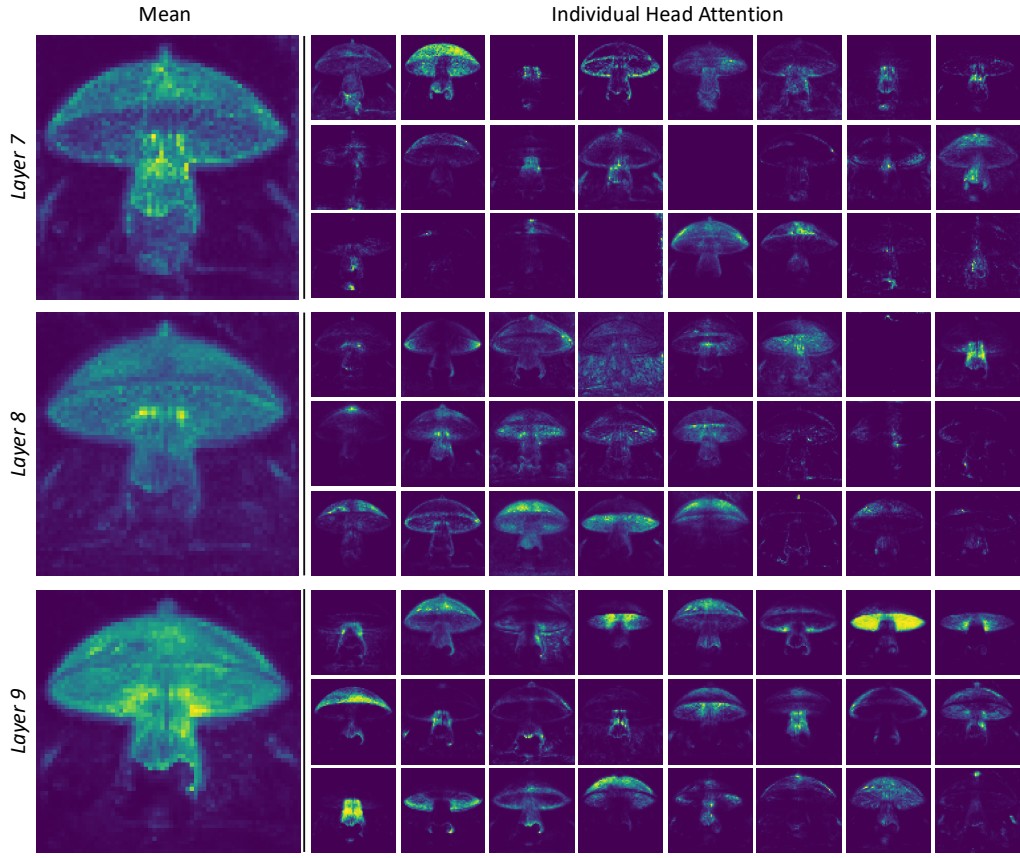

Figure 27: **Visualization of attention scores for all heads.**

## D.3 Statistics of attention score and norm

We present the statistics of L2 norms of value-projected features across all layers and heads in Fig.28. Notably, layer 9 exhibits consistently large norms for text tokens, suggesting that text features strongly dominate this layer. When restricting the analysis to the actual prompt tokens only, this trend becomes even more pronounced.

On the other hand, we observe large image token norms in specific heads within layers 7 and 8, which coincide with the heads that show strong segmentation performance in Fig. 17. This suggests that these heads are highly responsive to image-specific features and may play a crucial role in localizing visual semantics prior to cross-modal alignment with text.

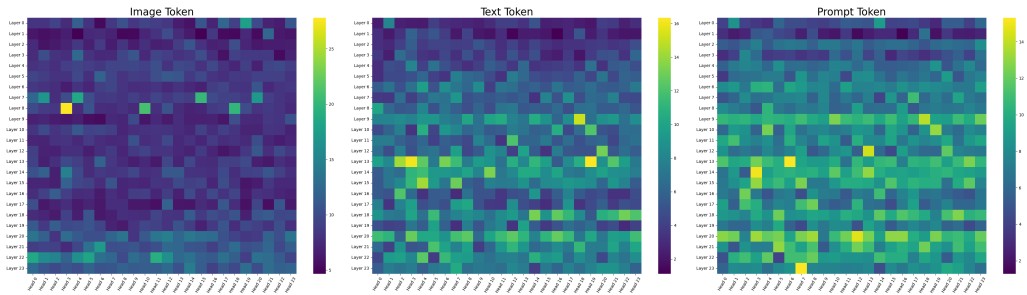

Figure 28: **Average L2 norm of value tokens by layer and head.**

## D.4 PCA visualization of attention features

We provide PCA visualizations of the query-, key-, and value-projected image features across all layers in Fig. 29. Most layers exhibit a strong positional bias, whereas layer 9 reveals distinct semantic grouping. This suggests that image features become semantically well-grounded at layer 9, enabling more meaningful cross-modal interactions.

## D.5 Emergent behavior of <pad> tokens

In Fig. 30, we visualize all 77 tokens under the unconditional generation setting described in Sec. 3.4, which includes <sos>, <eos>, and 75 <pad> tokens. Remarkably, we observe that individual <pad> tokens attend to distinct semantic regions, despite the absence of explicit semantic information in the text prompt.

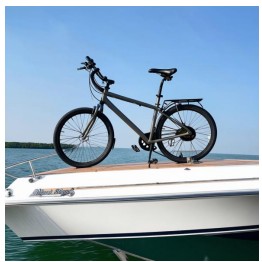
*"A bicycle on top of the boat"*

Figure 29: **PCA visualization of query-, key-, and value-projected feature.**

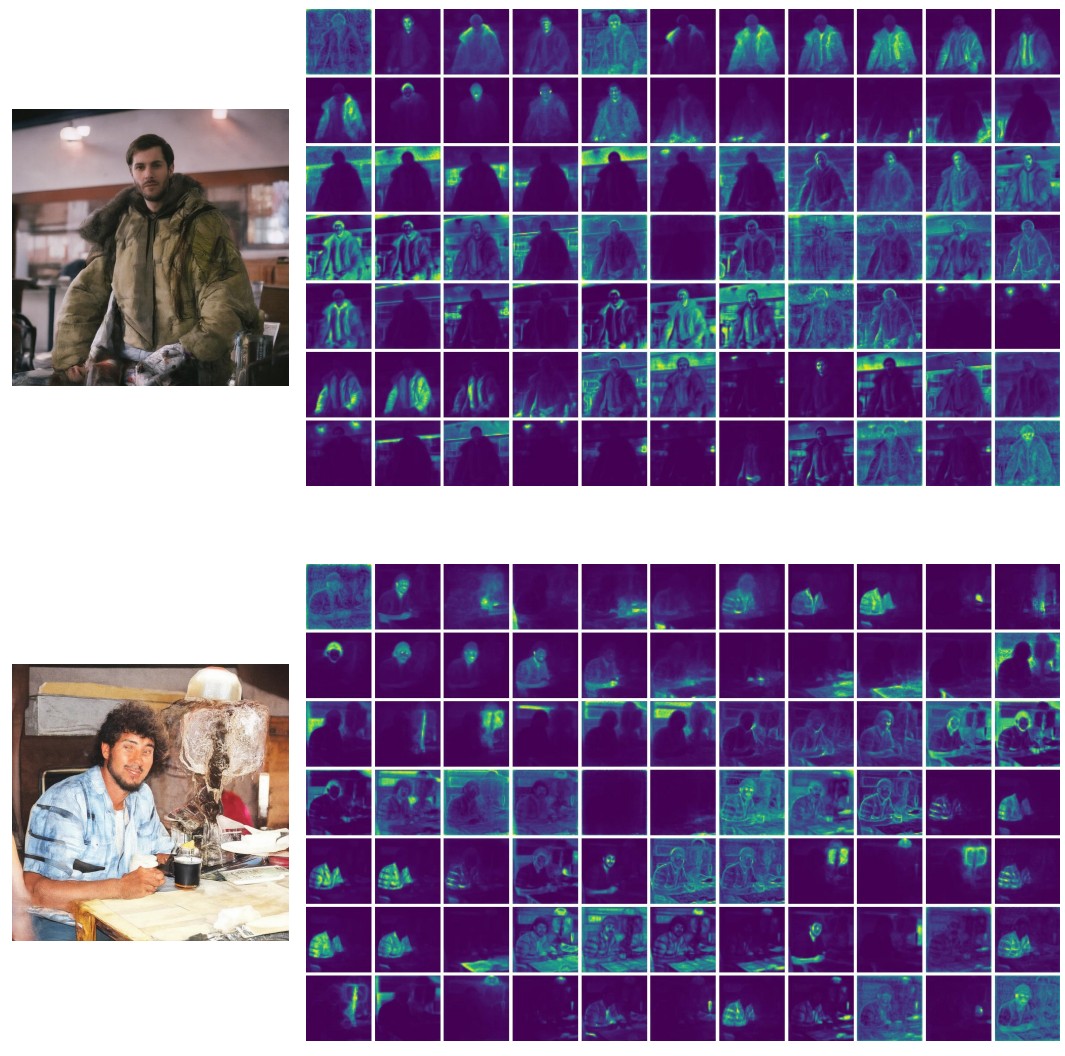

Figure 30: **Emergent behavior of <pad> tokens in unconditional generation.**

# E  Additional qualitative results for generation

We present extended results from attention perturbation experiments in Sec 3.3. Perturbations applied to layers other than layer 9 result in minor degradation of image fidelity or structure while largely preserving semantic content. In contrast, perturbing layer 9 leads to the generation of semantically irrelevant images. Conversely, if we leverage this perturbed sample as a negative sample for the guidance, we observe a substantial gain on the image quality. This strongly supports our claim that layer 9 plays a critical role in cross-modal interaction, with a particular emphasis on aligning with the text modality.

## E.1  Extended I2T attention perturbation results

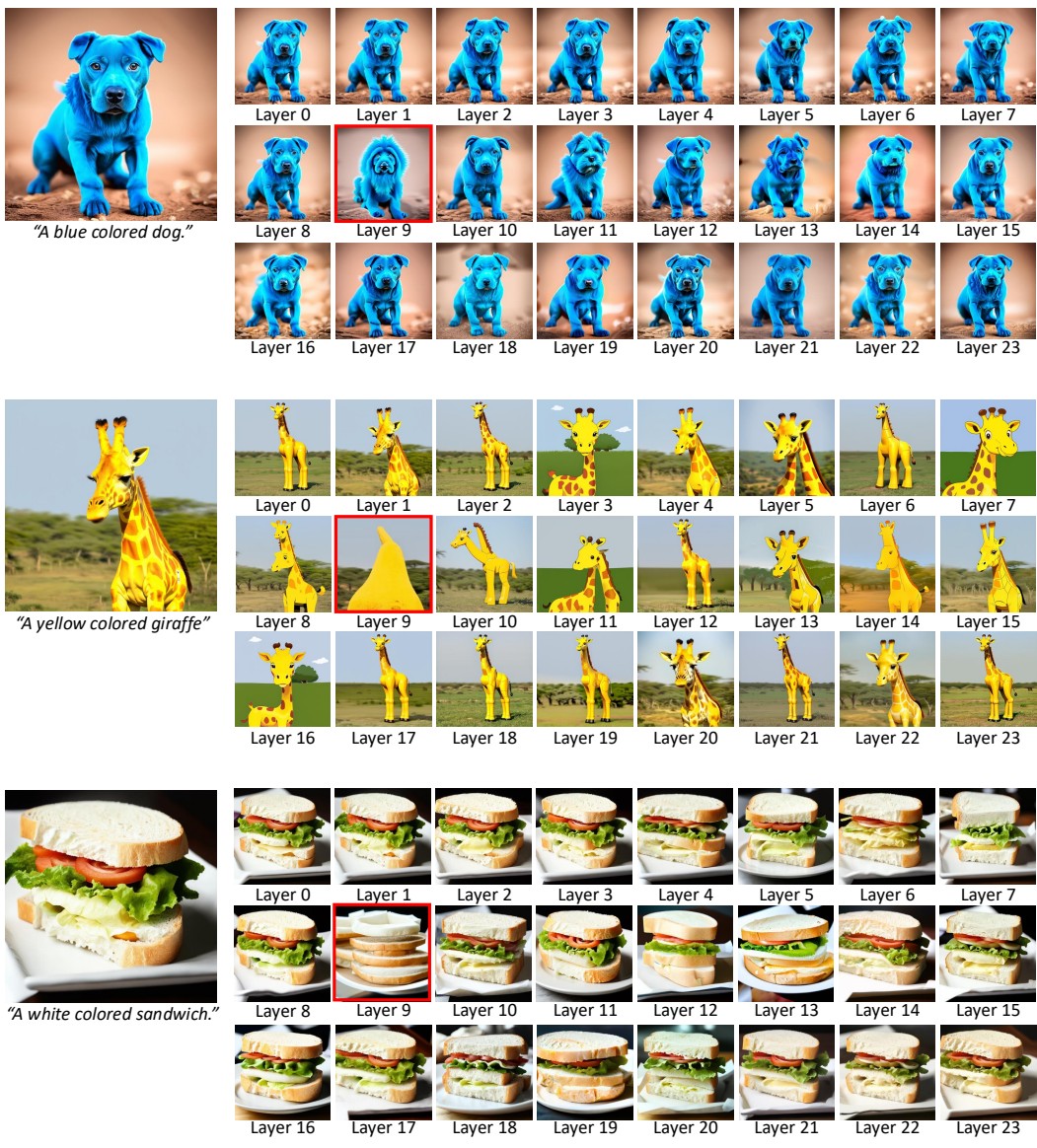

Figure 31: **Effect of applying attention perturbation at different layers during image generation.**

## E.2 Extended I2T attention perturbation guidance results

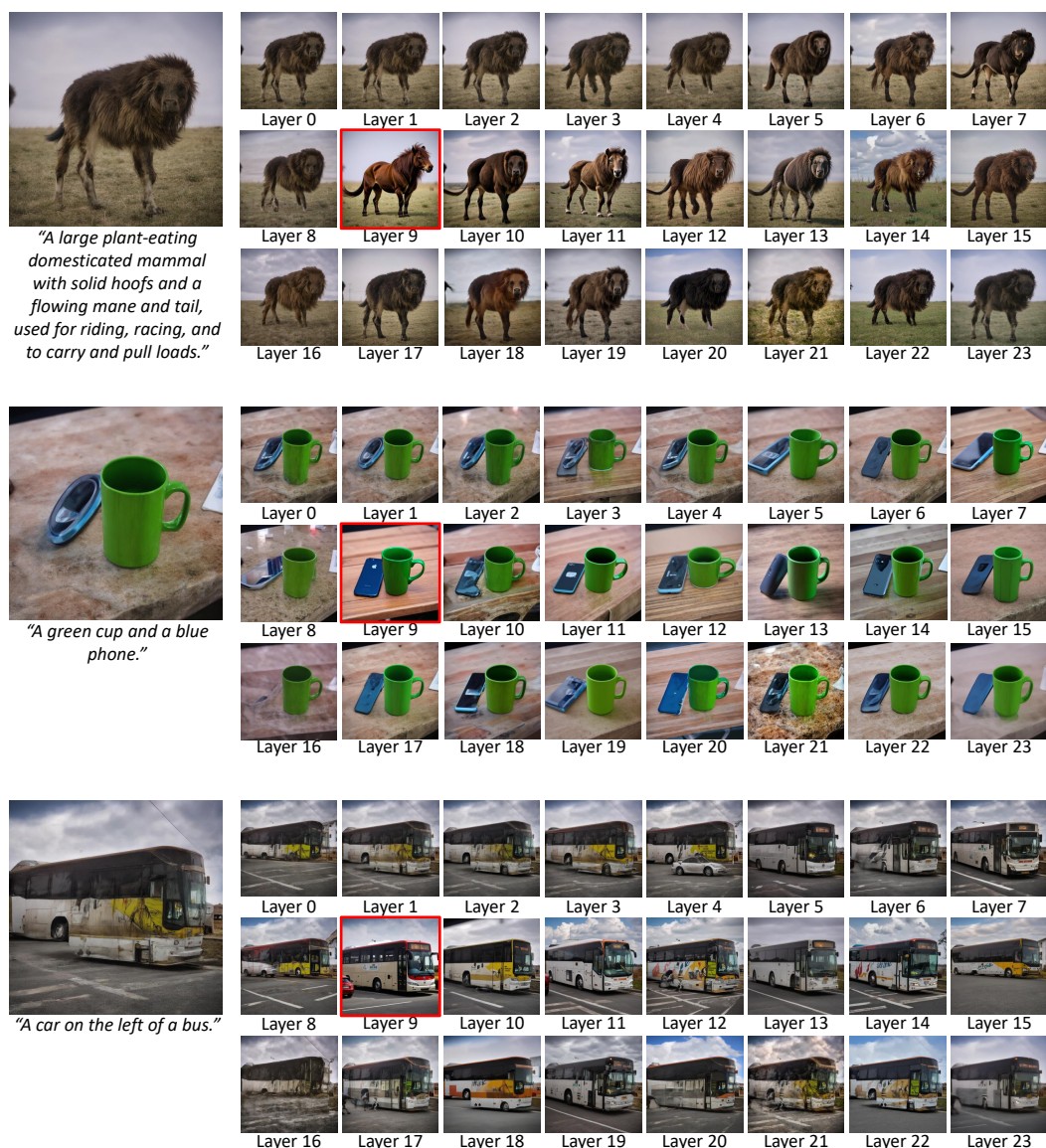

Figure 32: **Effect of applying attention perturbation guidance at different layers during image generation.**

## E.3 Image generation results of our trained models

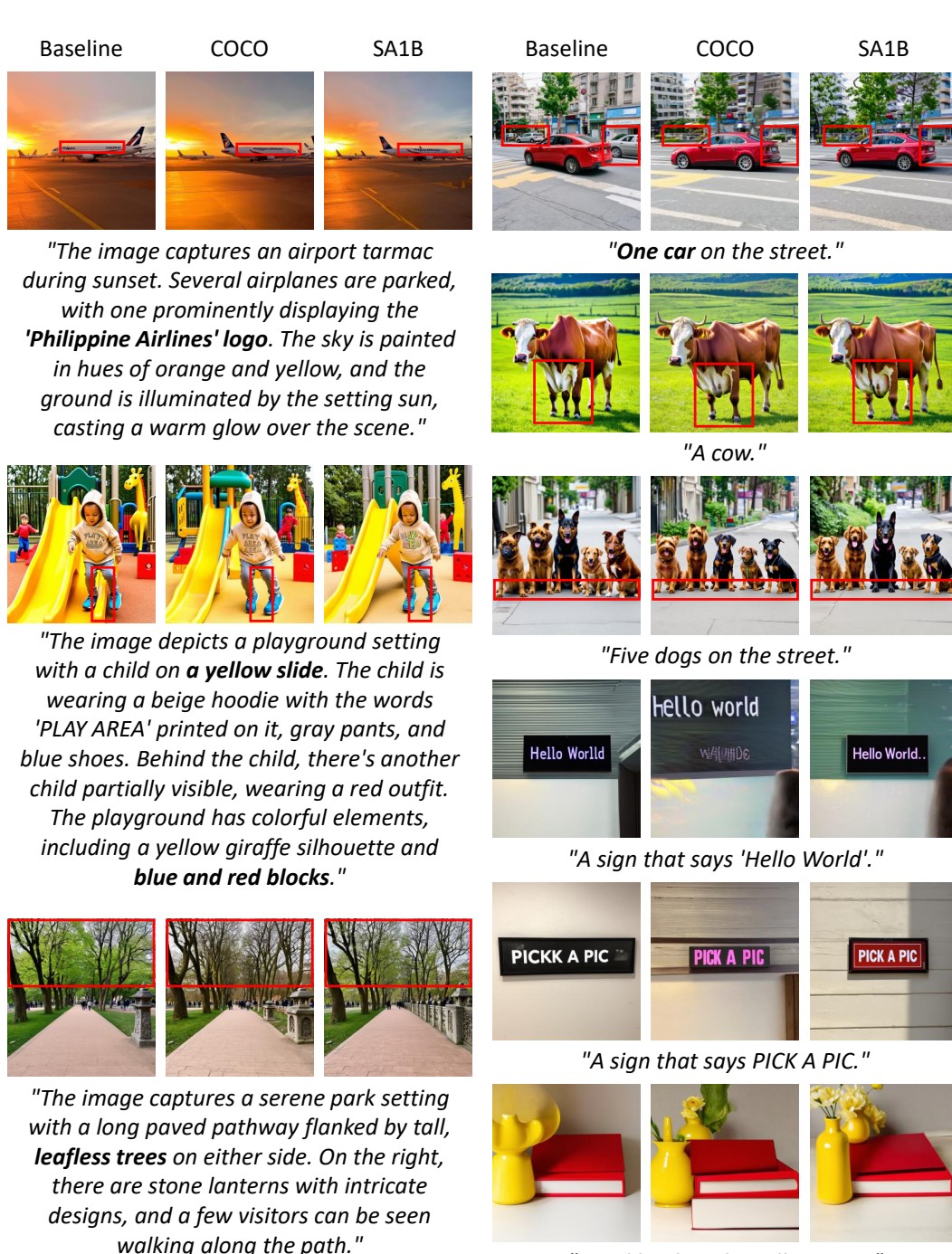

Figure 33: **Qualitative results of trained models.**

# F Additional qualitative results for segmentation

## F.1 Open-vocabulary semantic segmentation results

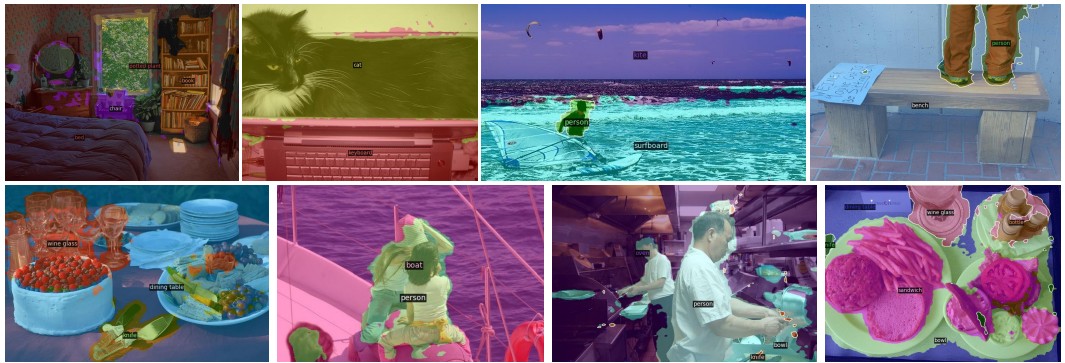

Figure 34: **Open-vocabulary semantic segmentation results of Seg4Diff on COCO-Object.**

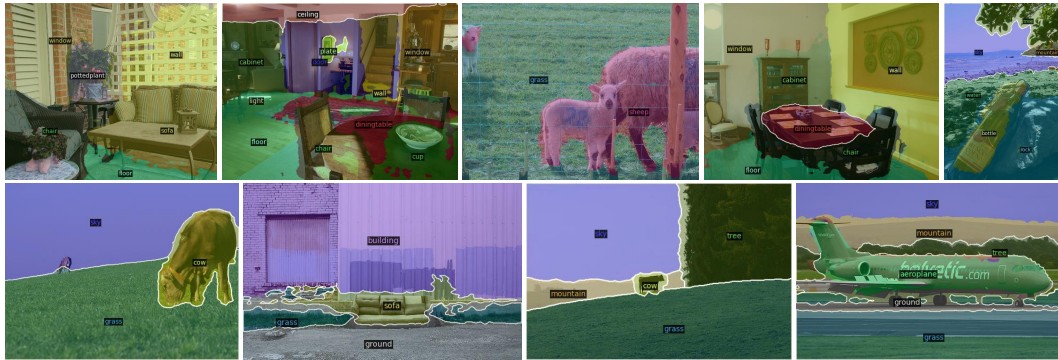

Figure 35: **Open-vocabulary semantic segmentation results of Seg4Diff on Pascal-Context59.**

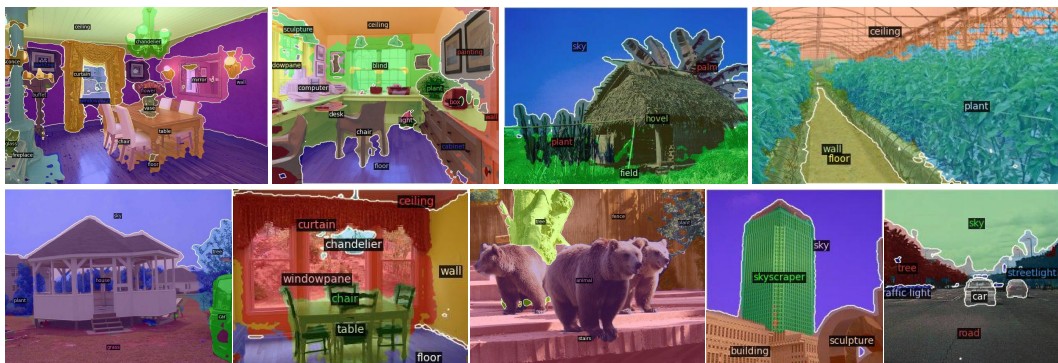

Figure 36: **Open-vocabulary semantic segmentation results of Seg4Diff on ADE20K.**

## F.2 Unsupervised segmentation results

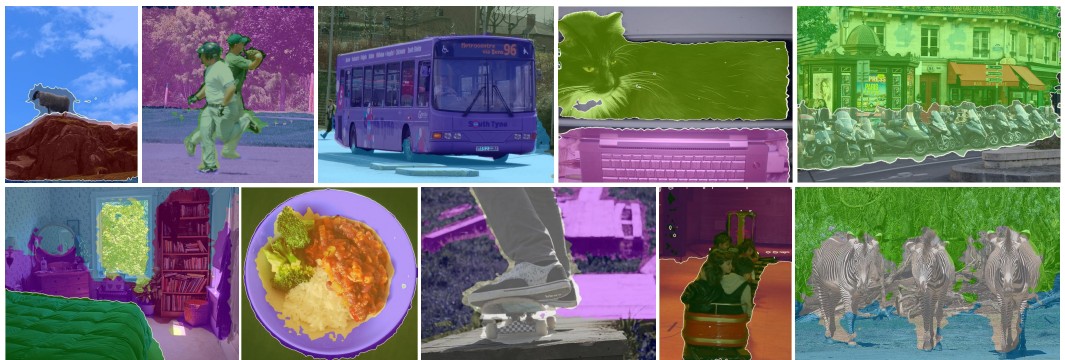

Figure 37: **Unsupervised segmentation results of Seg4Diff on COCO.**

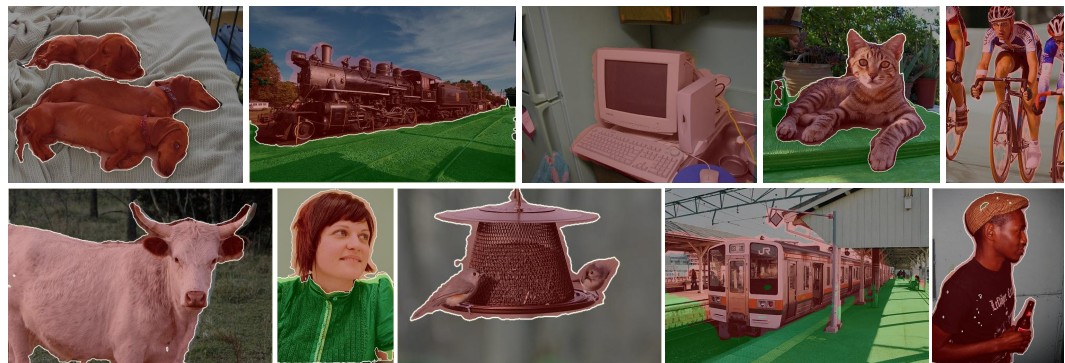

Figure 38: **Unsupervised segmentation results of Seg4Diff on VOC.**

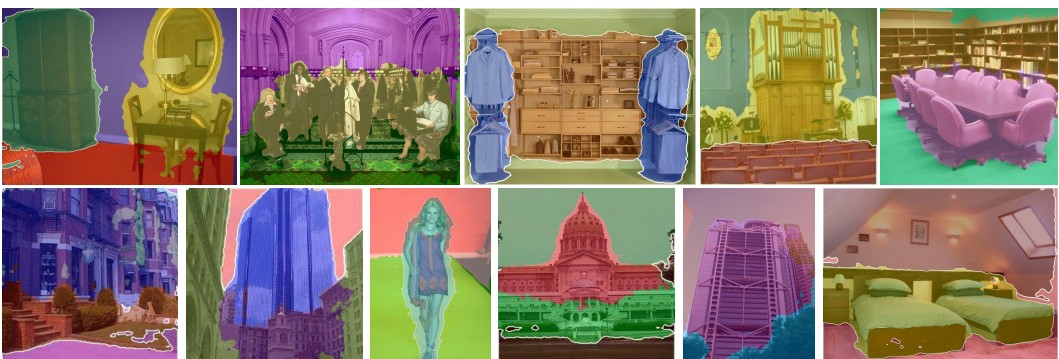

Figure 39: **Unsupervised segmentation results of Seg4Diff on ADE20K.**

# G   Limitations of our method

Our method evaluates segmentation without postprocessing or upsampling, which limits performance on extremely small objects. In addition, diffusion models may ground class names differently from the ground-truth annotations, introducing an inherent mismatch that impacts accuracy. Addressing this representation–annotation gap is an important direction for future work.

We focus on dense perception and image generation tasks, deferring reasoning-centric evaluations [65] (e.g., action recognition, spatial-relations QA) to future work, as our supervision targets segmentation semantics rather than high-level reasoning. For simplicity, the loss is applied to a single "sweet-spot" layer per backbone; broader layer/timestep exploration or auxiliary heads (e.g., optical flow, pose) could yield further gains without changing our core findings.

