# OpenReview forum: "Seg4Diff: Unveiling Open-Vocabulary Semantic Segmentation in Text-to-Image Diffusion Transformers"
_NeurIPS.cc/2025/Conference — NeurIPS 2025 poster_

### Official Review · Reviewer_tfvR · 2025-06-11

**Clarity:** 3
**Significance:** 2
**Originality:** 2
**Rating:** 4
**Confidence:** 4

**Summary:**

This paper investigates how semantic information is structured and propagated within Multi-Modal Diffusion Transformers (MM-DiTs) by analyzing their attention mechanisms. The authors identify specific attention heads and layers that align text tokens with spatially coherent image regions, contributing to both image generation and zero-shot segmentation. They propose a lightweight LoRA-based fine-tuning approach to enhance these capabilities without compromising image quality. The study reveals that semantic alignment is an emergent property of diffusion transformers and can be selectively strengthened to improve both generation and perception tasks.

**Questions:**

- Could you provide additional qualitative comparisons between the baseline and fine-tuned models in the generation setting to better illustrate semantic or visual differences?
- ZestGuide [3] proposes a mechanism that seems similar to the one introduced in Figure 6, albeit in a zero-shot conditional generation context. Could you clarify how your approach differs? Have you considered comparing against such methods? This could further support the findings in Table 3 and reinforce the claims about improved generation.
- Line 157 mentions reporting results at timesteps T=14 and T=28, but Appendix A.1 only refers to T=14 of 28. It is unclear where T=28 is presented. Have you conducted any ablation over timesteps to evaluate how semantic richness evolves? This could add additional insight to your analysis.

If the authors provide additional experiments, on other backbones and more benchmarks, I’ll be happy to raise my score.

Typos:
- L.171: now To &rarr; now to
- L.187 image-text alignment,, &rarr; image-text alignment,
- L.190 it can be observed that the only the layers &rarr; it can be observed that only the layers

[3] Couairon, G., Careil, M., Cord, M., Lathuiliere, S., & Verbeek, J. (2023). Zero-shot spatial layout conditioning for text-to-image diffusion models. In Proceedings of the IEEE/CVF International Conference on Computer Vision (pp. 2174-2183).

**Ethical Concerns:**

["NO or VERY MINOR ethics concerns only"]

**Final Justification:**

My initial concerns have been addressed in the rebuttal, this papers shows valuable insights into how semantic representations emerge and aggregate in DiT models, providing meaningful guidance toward unified models that bridge visual generation and perception. I increase my score to borderline accept.

**Limitations:**

I don't see a discussion about the limitations of the method. Could you discuss the limitations in the supplementary material as mentioned in the checklist.

**Paper Formatting Concerns:**

No major formatting issue.

**Quality:**

3

**Strengths And Weaknesses:**

**Strengths**
- The paper is clearly written and well-structured, making it easy to follow.
- It addresses an underexplored area—how semantic information is stored and propagated within MM-DiT models. This is a highly relevant topic, as understanding these mechanisms can aid in designing more effective diffusion transformers with improved image-text alignment, potentially boosting generative capabilities and downstream performance.
- Figures are well-designed and effectively convey the main findings.
- The experimental analysis is thoughtful and yields valuable insights into the internal behavior of these models.

**Weaknesses**
- The experiments are limited to Stable Diffusion 3, which restricts the generalizability of the findings. Expanding the evaluation to models like SD3.5 or FLUX, and including a range of model scales (as in Figures 1, 2, and 4), would help assess the robustness and scalability of the approach.
- The claim in Line 200 that the proposed method “ultimately boosts segmentation performance” is not fully supported by the results. Table 1 compares only two baselines, and Table 2 presents mixed outcomes. Although Diff4Seg shows improved mIoU, it performs worse in accuracy compared to DiffSeg on Cityscapes. The fine-tuned Diff4Seg shows only marginal gains—primarily on COCO-Stuff-27 mIoU. Additionally, DiffCut is mentioned in related work but not included in the comparison. Broader benchmarking (e.g., on VOC20, ADE20K) would strengthen this claim.
- Similarly, the claim of improved “generation performance” (Line 200) lacks convincing evidence. In Table 3, only the FID score improves; IS and CLIP Score do not outperform the baseline. While evaluating generation quality is inherently difficult, incorporating more metrics (e.g., FD-DINOv2 [1], Attribute Binding and Spatial Relationships [2] from T2I-CompBench) and more qualitative examples could better support the claim.

[1] Stein, G., Cresswell, J., Hosseinzadeh, R., Sui, Y., Ross, B., Villecroze, V., ... & Loaiza-Ganem, G. (2023). Exposing flaws of generative model evaluation metrics and their unfair treatment of diffusion models. Advances in Neural Information Processing Systems, 36, 3732-3784.

[2] Huang, K., Sun, K., Xie, E., Li, Z., & Liu, X. (2023). T2i-compbench: A comprehensive benchmark for open-world compositional text-to-image generation. Advances in Neural Information Processing Systems, 36, 78723-78747.

---

> ### Author Rebuttal · Authors · 2025-07-31
>
> ### **Generalizability to different models**
> Due to the character limit, we kindly refer the reviewer to the "Generalizability to different models" in the rebuttal for revewer FeAF. We thank the reviewer for the understanding.
>
> ### **Clarification on expressions**
>
> We greatly appreciate the reviewer’s constructive suggestion. Below, we have conducted broader benchmarking on unsupervised segmentation settings and compared it with the suggested baseline. In addition to the SA-1B–trained model presented in the manuscript, we also include COCO-trained variants for comparison. The results are summarized below:
>
> | Model | Training dataset | VOC20 | PC59 | COCO-Obj | COCO-Stuff-27 | Cityscapes | ADE20K |
> | --- | --- | --- | --- | --- | --- | --- | --- |
> | DiffCut [1] | - | 62.0 | 54.1 | 32.0 | 46.1 | 28.4 | 42.4 |
> | Ours (Zero-shot) | - | 54.9 | 52.6 | 38.5 | 49.7 | 24.2 | 44.9 |
> | Ours (Trained) | SA1B | 55.1 (+0.2) | 52.8 (+0.2) | 39.0 (+0.5) | 50.8 (+1.1) | 24.2 (0.0) | 45.0(+0.1) |
> | Ours (Trained) | COCO | 56.1 (+1.2) | 53.5 (+0.9) | 38.8 (+0.3) | 53.5 (+3.8) | 24.4 (+0.2) | 45.4 (+0.5) |
>
> These results confirm that the insights from our layer-selection analysis translate into tangible improvements across multiple datasets, and they also invite future work on more sophisticated supervisory designs.
>
> In light of the reviewer’s comment, we will replace “*ultimately boosts both segmentation and generation performance*” with a more precise statement: *“The proposed supervision improves segmentation accuracy without degrading, and in some cases modestly enhances, image generation quality.”* We apologise for any earlier overstatement.
>
> [1] Couairon, Paul, et al. "Diffcut: Catalyzing zero-shot semantic segmentation with diffusion features and recursive normalized cut." Advances in Neural Information Processing Systems 37 (2024): 13548-13578.
>
> ### **Regarding improvement in generation quality**
>
> We evaluate the CLIP score to assess the alignment between generated images and their corresponding text prompts. To ensure robustness against randomness, we conduct evaluations on three diverse prompt sets: (1) 500 prompts from Pick-a-Pic [2], with five images generated per prompt; (2) 5,000 images generated using prompts from the COCO caption validation set; and (3) captions generated by CogVLM for 1,000 validation images sampled from SA-1B, which do not overlap with our training data.
>
> | **clipscore ↑** | **Training dataset** | **Pick-a-Pic** | **COCO** | **SA-1B** | **Mean** |
> | --- | --- | --- | --- | --- | --- |
> | **Baseline** | - | 27.0252 | 26.0638 | 28.3422 | 27.1437 |
> | **Ours (Trained)** | SA1B | **27.0547** | 26.2318 | 28.4476 | 27.2447 |
> | **Ours (Trained)** | COCO | 27.0409 | **26.2319** | **28.5553** | **27.2760** |
>
> Based on the reviewer’s feedback, we also evaluated our model on T2I-Compbench++ [3].
>
> | **T2I compbench++** | **Training dataset** | **color** | **shape** | **texture** | **2D spatial** | **3D spatial** | **numeracy** | **non spatial** | **complex** |
> | --- | --- | --- | --- | --- | --- | --- | --- | --- | --- |
> | **Baseline** | **-** | 0.7864 | 0.5644 | 0.7200 | **0.2435** | **0.3318** | 0.5566 | **0.3124** | 0.3719 |
> | **Ours (Trained)** | **SA1B** | 0.7836 | 0.5679 | 0.7252 | 0.2330 | 0.3151 | 0.5460 | 0.3113 | 0.3709 |
> | **Ours (Trained)** | **COCO** | **0.7919** | **0.5687** | **0.7260** | 0.2301 | 0.3234 | **0.5584** | 0.3120 | **0.3735** |
>
> Overall, our method outperforms the baseline on nearly every metric. The only exceptions are the reasoning-focused benchmarks—specifically the 2-D/3-D spatial and non-spatial tests—where our scores trail the baseline. This shortfall is unsurprising: those tasks demand explicit spatial reasoning and action understanding, whereas our supervisory signal was designed to strengthen perceptual and segmentation cues, not high-level reasoning. In light of the reviewer’s comment, we will also rephrase the claims by slightly toning down, if the reviewer agrees. We apologise for any earlier overstatement.
>
> Also, we will conduct and include a user study to better assess the generation quality of ours in the final version.
>
> [2] Kirstain, Yuval, et al. "Pick-a-pic: An open dataset of user preferences for text-to-image generation." Advances in neural information processing systems 36 (2023): 36652-36663.
>
> [3] Huang, Kaiyi, et al. "T2i-compbench: A comprehensive benchmark for open-world compositional text-to-image generation." Advances in Neural Information Processing Systems 36 (2023): 78723-78747.
>
> ### **Comparison with ZestGuide**
> We thank the reviewer for suggesting discussions regarding ZestGuide [4]. While both ZestGuide and ours leverage similar mask loss design, we notice that as ZestGuide targets layout generation, its goal is to enforce the diffusion model to generate the instance in the mask area. In contrast, our motivation is to improve multi-modal interaction within certain layers, rather than enforcing certain objects to be grounded to certain locations. This also differentiates our work with ZestGuide in terms of training configuration, where we generally fine-tune the model in a set of images, whereas ZestGuide requires per-image optimization. Nonetheless, we appreciate the reviewer’s pointer and remain open to further suggestions.
>
> [4] Couairon, Guillaume, et al. "Zero-shot spatial layout conditioning for text-to-image diffusion models." Proceedings of the IEEE/CVF International Conference on Computer Vision. 2023.
>
> ### **Regarding the choice of timestep**
>
> We apologize for the typo in L157: the text should read “**T = 14 out of 28**”, not “T = 14 and T = 28”. We will correct this in the revision.
>
> Initially, we chose **t=14** as our early experiments on ADE20K showed the best result for t=14 when we conducted a coarse search over the timesteps. To further explore the effects of timesteps, we evaluate over **all** 28 timesteps in VOC and COCO-Object, and present the results below. Note that we only show some of the best and worst performing layers for brevity.
>
> | **Dataset \ Timestep** | **1** | **2** | **6** | **7** | **8** | **14** | **21** | **28** |
> | --- | --- | --- | --- | --- | --- | --- | --- | --- |
> | **Pascal VOC** | 73.9 | 84.5 | **89.2** | 89.0 | **89.2** | 87.8 | 81.0 | 70.5 |
> | **COCO-Object** | 43.9 | 55.4 | 62.0 | **62.1** | **62.1** | 58.2 | 48.5 | 25.4 |
>
> The results reveal that t=8 in fact consistently outperforms our original choice of t=14 for both VOC and COCO-Object, with t=6 and 7 similarly showing strong results. While we default the timestep to t=14 for additional experiments presented in the rebuttals to avoid confusion, we thank the reviewers for pointing this out and promise to update the scores as well as include the ablation on timesteps in the revised manuscript.
>
> ### **Typos**
>
> Thank you for pointing out the typos. We apologize for any confusion caused by this and will update our manuscript accordingly.
>
> ### **Limitations of our method**
>
> We apologize for not including the limitations of our method in the supplementary material. To address this oversight, we provide the limitations below, which will certainly be included in our revised version.
>
> Our study deliberately focuses on dense perception and image‐generation tasks, leaving reasoning-centric evaluations (e.g., action recognition or spatial-relations QA) to future work because our supervision signal targets segmentation semantics rather than high-level reasoning. In addition, we attach the loss to a single “sweet-spot” layer per backbone for maximal simplicity; while this already delivers strong gains, a more exhaustive layer/timestep sweep—or the addition of auxiliary heads such as optical flow or pose—could yield incremental improvements without altering our core conclusions.

---

> > ### Comment · Reviewer_tfvR · 2025-08-04
> >
> > I appreciate the additional material provided in the rebuttal and acknowledge the authors’ efforts in addressing the reviewers’ concerns. The paper offers valuable insights into how semantic representations emerge and aggregate in DiT models, providing meaningful guidance toward unified models that bridge visual generation and perception. The explanations in the rebuttal have addressed my initial concerns, and I will therefore raise my score to a borderline accept. I strongly encourage the authors to incorporate the supplementary quantitative results shared in the rebuttal into the main paper for completeness and clarity.

---

> > > ### Author Response · Authors · 2025-08-05
> > >
> > > Thank you for your thoughtful review and for taking the time to engage with our work. We truly appreciate your response and feedback.

---

### Official Review · Reviewer_3ee8 · 2025-06-16

**Clarity:** 3
**Significance:** 2
**Originality:** 2
**Rating:** 4
**Confidence:** 4

**Summary:**

This paper explores the attention mechanism of MM-DiT in diffusion models, focusing on how specific heads and layers propagate semantic information and influence generation quality. This paper identifies critical layers in MM-DiT that reveal strong influences on image segmentation and generation. A corresponding fine-tuning-based and training-free method is proposed for enhancing the segmentation and generation qualities.

**Questions:**

See Weaknesses

**Ethical Concerns:**

["NO or VERY MINOR ethics concerns only"]

**Final Justification:**

I think the main issues have been addressed, and the update score is 4. Since the method's sensitivity to targets of varying sizes is unclear, the authors should discuss these limitations in the final version and provide the necessary numerical results.

**Limitations:**

yes

**Quality:**

2

**Strengths And Weaknesses:**

Strengths:

1. The insight into how attention maps influence segmentation and generation is interesting.

2. The writing style is easy to follow.

Weaknesses:

1. This paper discusses the attention map of the MMdit architecture but is limited to SD3. The authors are encouraged to explore MM-DiT-based models, such as FLUX [1], to demonstrate the generalizability of the findings.

2. There have been some works discussing the application of mutual attention maps and self-attention maps in image segmentation. I am confused about the difference between this paper and previous works[2,3]. In addition, in Tab.2, the performance of Diff4Seg is significantly lower than [2].

3. Some expressions are not unified. For example, L263 uses the abbreviation `Tab.2`, but L250 uses the full name `Table 1`.

[1] https://github.com/black-forest-labs/flux

[2] Diffuse attend and segment: Unsupervised zero-shot segmentation using stable diffusion. CVPR, 2024.

[3] Diffusion model is secretly a training-free open vocabulary semantic segmenter. TIP, 2025.

---

> ### Author Rebuttal · Authors · 2025-07-31
>
> ### **Generalizability to different models**
>
> To demonstrate that our insights are not tied to SD3, we replicated the entire analysis on two additional MM‑DiT variants—SD 3.5 [1] and Flux [2]. Summarizing the results, we observed consistent trends across all models (Note that the exact layer‑wise statistics can differ between models).
>
> Because the rebuttal cannot include external media, we present the key quantitative results here and will supply the corresponding plots and qualitative examples in the camera‑ready version. Following the methodology in the main paper, we use three complementary evaluation protocols:
>
> - **Attention‑norm analysis** – We compute the value‑vector norms for image, text, and actual prompt tokens at every layer, highlighting only the most informative layers for readability.
> - **Attention‑perturbation analysis** – For each layer, we perturb its cross‑attention and measure DINO, CLIP‑I, and CLIP‑T similarities between the original and perturbed images; larger drops indicate layers that are critical to image fidelity.
> - **Segmentation evaluation** – Using cross‑attention maps as pseudo‑masks, we evaluate mIoU on Pascal VOC to assess mask quality.
>
> In the table, bold indicates the highest entry for value norm and segmentation metrics, and the lowest for perturbation metrics.
>
> **Stable Diffusion 3.5 Medium**
>
> | Layer | 3 | 7 | 8 | 9 | 12 | 13 | 15 | 16 |
> | --- | --- | --- | --- | --- | --- | --- | --- | --- |
> | ***[Value norm ratio]*** |  |  |  |  |  |  |  |  |
> | Image tokens | **48.5** | 39.5 | 39.1 | 27.7 | 35.8 | 32.1 | 37.3 | 42.8 |
> | Text tokens | 17.4 | 29.3 | 29.8 | **41.5** | 27.4 | 41.2 | 33.3 | 23.2 |
> | Prompt tokens | 22.9 | 35.6 | 37.3 | **53.8** | 37.0 | 47.2 | 44.8 | 32.0 |
> | ***[Perturbation metric]*** |  |  |  |  |  |  |  |  |
> | DINO | 0.8910 | 0.7814 | 0.8314 | **0.6011** | 0.8432 | 0.7929 | 0.8507 | 0.9056 |
> | CLIP-I | 0.9511 | 0.9232 | 0.9333 | **0.8701** | 0.9407 | 0.9319 | 0.9434 | 0.9599 |
> | CLIP-T | 0.3188 | 0.3194 | 0.3181 | **0.3108** | 0.3182 | 0.3177 | 0.3151 | 0.3175 |
> | Harmonic mean | 0.5649 | 0.5460 | 0.5537 | **0.4974** | 0.5564 | 0.5472 | 0.5546 | 0.5665 |
> | ***[Semseg eval]*** |  |  |  |  |  |  |  |  |
> | PascalVOC | 72.6 | 82.8 | 82.8 | **84.7** | 83.4 | 82.9 | 78.3 | 75.3 |
>
> **Flux.1-dev**
>
> | Layer | 0 | 2 | 6 | 8 | 9 | 10 | 12 | 15 | 17 | 26 |
> | --- | --- | --- | --- | --- | --- | --- | --- | --- | --- | --- |
> | ***[Value norm]*** |  |  |  |  |  |  |  |  |  |  |
> | Image tokens | 57.1 | 85.8 | 41.9 | 54.8 | 52.8 | 53.6 | 56.7 | 88.9 | **89.3** | 70.6 |
> | Text tokens | 63.4 | **91.9** | 69.8 | 70.8 | 67.9 | 74.1 | 66.3 | 58.0 | 70.1 | 56.7 |
> | Prompt tokens | 72.4 | 106.3 | 104.4 | 115.1 | 103.8 | 120.6 | 127.1 | 108.6 | **139.8** | 57.3 |
> | ***[Perturbation metric]*** |  |  |  |  |  |  |  |  |  |  |
> | DINO | 0.9737 | 0.9922 | 0.9947 | 0.9829 | 0.9780 | 0.9635 | **0.9524** | 0.9820 | 0.9617 | 0.9699 |
> | CLIP-I | 0.9899 | 0.9972 | 0.9978 | 0.9938 | 0.9906 | 0.9881 | **0.9832** | 0.9946 | 0.9896 | 0.9920 |
> | CLIP-T | 0.3022 | 0.3038 | 0.3029 | 0.3031 | 0.3039 | 0.3040 | **0.3013** | 0.3036 | 0.3044 | 0.3033 |
> | Harmonic mean | 0.5611 | 0.5657 | 0.5650 | 0.5636 | 0.5636 | 0.5618 | 0.5569 | 0.5641 | **0.5546** | 0.5622 |
> | ***[Semseg eval]*** |  |  |  |  |  |  |  |  |  |  |
> | PascalVOC | 65.9 | 68.0 | 70.0 | 73.8 | 73.3 | 71.6 | 79.7 | 74.5 | **83.5** | 77.0 |
>
> From the table above, we can similarly observe the correlation between value norm and segmentation performance among layers for both Flux and Stable Diffusion 3.5. While SD3.5 appears to have layer 9, identical to SD3, to exhibit strong semantic grounding, Flux shows layer 12 and 17 to have a similar tendency. This hints that our observation and methodology are not proprietary for SD3, but can be applied to other DiT-based diffusion models with multi-modal attention, highlighting generalizability of our insights and findings.
>
>
> ### **Comparison with previous attention-based methods**
>
> While we acknowledge that performing segmentation based on attention maps of diffusion models is an existing idea, we highlight that our exploration mainly focuses on understanding MM-DiT-based diffusion models and explore emergent properties and reveal that certain layers are able to show strong segmentation capabilities. On the other hand, [1,2] focuses on gathering information within the attention maps to identify segments, without fully regarding the image-text interaction within the diffusion models. This requires iterative algorithms [1] or tuning highly sensitive hyperparameters [2], whereas our method can simply obtain segmentation directly from a single attention layer and show competitive results. We also highlight that despite being simple, we outperform [2] in terms of IoU, while having a deficit in accuracy.
>
> We wish to highlight that our key contribution lies more on the in-depth analysis and investigation of learned representations of MM-DiT, exploration of critical lays for segmentation capabilities, and demonstration of strong zero-shot segmentation performance with further possible enhancement using localization capabilities of the critical layers.
>
> [1] Tian, Junjiao, et al. "Diffuse attend and segment: Unsupervised zero-shot segmentation using stable diffusion." Proceedings of the IEEE/CVF Conference on Computer Vision and Pattern Recognition. 2024.
>
> [2] Wang, Jinglong, et al. "Diffusion model is secretly a training-free open vocabulary semantic segmenter." IEEE Transactions on Image Processing (2025).
>
> ### **Unified expressions**
>
> Thank you for pointing out the inconsistency in expressions. We will update our manuscript to unify those accordingly.
>
> ### **Limitations of our method**
>
> We apologize for not including the limitations of our method in the supplementary material. To address this oversight, we provide the limitations below, which will certainly be included in our revised version.
>
> Our study deliberately focuses on dense perception and image‐generation tasks, leaving reasoning-centric evaluations (e.g., action recognition or spatial-relations QA) to future work because our supervision signal targets segmentation semantics rather than high-level reasoning. In addition, we attach the loss to a single “sweet-spot” layer per backbone for maximal simplicity; while this already delivers strong gains, a more exhaustive layer/timestep sweep—or the addition of auxiliary heads such as optical flow or pose—could yield incremental improvements without altering our core conclusions.

---

> > ### Author Response · Authors · 2025-08-05
> >
> > Thank you again for the insightful feedback. As the discussion period has been extended, we are happy to continue the dialogue and address any remaining concerns. Please let us know if there is anything further we can clarify, and we will respond promptly.

---

> > ### Comment · Reviewer_3ee8 · 2025-08-06
> >
> > Thanks for the detailed response. Most of my concerns are addressed. However, I'm curious about the sensitivity of the proposed method to target size. For example, does it still work for very small objects? Could you discuss this?

---

> > > ### Author Response · Authors · 2025-08-06
> > >
> > > Thank you for your thoughtful follow-up. We agree that the target size can influence segmentation performance, especially in the case of very small objects.
> > >
> > > As our method directly leverages the attention map from the diffusion transformer (at 1/16 resolution) without additional upsampling or refinement modules, there can indeed be limitations in accurately segmenting extremely small objects. However, this challenge is not unique to our approach—most existing methods using encoder-based features (including U-Net-based baselines) also struggle with small target regions due to downsampling in early layers.
> > >
> > > Notably, our experiments suggest that the DiT attention maps offer sharper and more localized semantic correspondence than other baselines, which partially mitigates the issue. Moreover, established techniques such as multi-scale inference or zoom-in refinement could be integrated to further improve performance on small targets, and we consider this a valuable direction for future work.

---

> > > > ### Comment · Reviewer_3ee8 · 2025-08-09
> > > >
> > > > I think the main issues have been addressed, and the update score is 4. Since the method's sensitivity to targets of varying sizes is unclear, please discuss these limitations in the final version and provide the necessary numerical results.

---

> > > > > ### Author Response · Authors · 2025-08-09
> > > > >
> > > > > Thank you for your thoughtful review and for taking the time to engage with our work. We truly appreciate your response and feedback. We will include this discussion in our revision accordingly.

---

> > > ### Author Response · Authors · 2025-08-08
> > >
> > > Thank you again for the insightful feedback. As the discussion period is ending soon, we are happy to continue the dialogue and address any remaining concerns. We would appreciate hearing whether your concern regarding the target size has been resolved, and if there is anything further we can clarify, we will respond promptly.

---

### Official Review · Reviewer_wi3R · 2025-06-17

**Clarity:** 2
**Significance:** 2
**Originality:** 2
**Rating:** 4
**Confidence:** 4

**Summary:**

The manuscript analyse attention maps of MM-DiT, focusing on the alignment between semantic information in images and text tokens. From the analysis the authors identity the 9th layer to have higher semantic relevance, and to align text and images. Following this study, the authors inject LoRA layer in the network to improve adherence to the prompt. These layers are fine-tuned using the flow matching loss and a mask loss that encourages the alignment and segmentation text-image.

The results show that the alignment is an emergent property in these models, and the proposed approach has comparable results with baselines.

**Questions:**

1. Intuitively, do you expect these conclusion to be general to other MM-DiT models? (of course the layers may be different, it might be 9th and 15th)
2. Do you expect that such a supervision would benefit a model trained from scratch? There would be no need of LoRAs and we would have to manually select layers instead of a study
3. Given that perturbing some attention maps (Sec 3.4) does not have a major impact in the results, have the authors thought of removing I2T attentions for those layers?
4. How would the method compare (quantitatively) with non-diffusion approaches? Did the authors run any preliminary experiments in the early stage of the project?

**Ethical Concerns:**

["NO or VERY MINOR ethics concerns only"]

**Final Justification:**

The authors addressed all my concerns, and I am convinced there is merit in this analysis

**Limitations:**

The authors fail to adequately highlight limitations and future work of the proposed work.

**Paper Formatting Concerns:**

No issues

**Quality:**

2

**Strengths And Weaknesses:**

Strengths:
- the quantitative results show competitive/minimal drop performance wrt the selected baselines
- the qualitative results show better alignment and accurate generation than baselines
- the analysis is valuable and can help designing better and more efficient DiT models

Weaknesses:
- ground truth masks are necessary, which restricts the applicability of these methods
- the need of a diffusion model able to segment images seems limited
- the manuscript lacks strong baselines, such as "Segment Anything", "Diffuse, Attend, and Segment", or in general non-diffusion baselines (other examples [1,2])

[1] ProxyCLIP: Proxy Attention Improves CLIP for Open-Vocabulary Segmentation
[2] CorrCLIP: Reconstructing Correlations in CLIP with Off-the-Shelf Foundation Models for Open-Vocabulary Semantic Segmentation

---

> ### Author Rebuttal · Authors · 2025-07-31
>
> ### **Generalizability to different models**
>
> Due to the character limit, we kindly refer the reviewer to the "Generalizability to different models" in the rebuttal for reviewer FeAF. We thank the reviewer for the understanding.
>
> ### **Clarification on the method and the task**
> Our framework offers both a training-free path and a fine-tuning stage, neither of which **do not require ground-truth masks**—at training or inference. This lets us evaluate in zero-shot and open-vocabulary settings without sacrificing scope. When adaptation is desired, we fine-tune on pseudo-labels generated automatically by Segment Anything (SAM), so the method remains label-free and readily transferable to new domains and datasets.
>
> Extracting high-quality segments directly from diffusion models is particularly valuable: it turns the creation of otherwise costly segmentation labels into a by-product of sampling foundation models. Our study therefore focuses on uncovering the emergent segmentation behavior of MM-DiT diffusion models and on showing how a lightweight fine-tune can efficiently harness these properties for downstream tasks.
>
> ### **Comparison with more baselines**
>
> We thank the reviewer for emphasizing the need to include strong baselines in our comparison. Below, we clarify our evaluation choices:
>
> **Segment Anything (SAM)**
> We did not include SAM [1] because its prompt-driven, open-set mask-generation paradigm is fundamentally different from our class-conditional diffusion approach and thus out of scope as a peer baseline. Instead, SAM serves as an *upper bound* on achievable mask quality.
>
> **Diffuse, Attend, and Segment**
> We kindly invite the reviewer to **revisit Table 2**, where the “Diffuse, Attend and Segment [2]” is **already one of our core comparators**: its performance appears as a name of “DiffSeg”.
>
> **CLIP-based non-diffusion methods**
> We would like to highlight that **CLIP-based methods are not fair to compare with ours**, where CLIP-based evaluations [3,4] assume *no* ground-truth class labels given, whereas diffusion-based methods are evaluated *with* ground-truth class names provided. This difference in setting comes from the inherent characteristics of CLIP and diffusion models, where CLIP needs distractors, while diffusion models need highly descriptive information regarding the image.
>
> Although this setting mismatch precludes a perfectly fair head-to-head comparison, we provide a comparison below:
>
> | **Dataset** | **VOC20** | **Object** | **PC59** | **ADE** | **City** |
> | --- | --- | --- | --- | --- | --- |
> | ***[CLIP-based methods]*** |  |  |  |  |  |
> | ProxyCLIP | 83.3 | 49.8 | 39.6 | 24.2 | 42.0 |
> | CorrCLIP | 91.8 | 52.7 | 47.9 | 28.8 | 49.9 |
> | ***[Diffusion-based methods]*** |  |  |  |  |  |
> | DiffSegmenter | 82.6 | - | - | - | - |
> | iSeg | 82.9 | 57.3 | - | - | - |
> | Ours (Zero-shot, t=14) | 86.4 | 58.0 | 46.1 | 31.9 | 19.0 |
> | Ours (Zero-shot, t=8) | 89.2 | 62.0 | 49.0 | 34.2 | 26.5 |
>
> [1] Kirillov, Alexander, et al. "Segment anything." Proceedings of the IEEE/CVF international conference on computer vision. 2023.
>
> [2] Tian, Junjiao, et al. "Diffuse attend and segment: Unsupervised zero-shot segmentation using stable diffusion." Proceedings of the IEEE/CVF Conference on Computer Vision and Pattern Recognition. 2024.
>
> [3] Lan, Mengcheng, et al. "Proxyclip: Proxy attention improves clip for open-vocabulary segmentation." European Conference on Computer Vision. Cham: Springer Nature Switzerland, 2024.
>
> [4] Zhang, Dengke, Fagui Liu, and Quan Tang. "Corrclip: Reconstructing correlations in clip with off-the-shelf foundation models for open-vocabulary semantic segmentation." arXiv preprint arXiv:2411.10086 (2024).
>
> ### **Benefit of our method for from-scratch training**
>
> We thank the reviewer for suggesting a valuable point for discussion. Although we can not afford the computational budget to train a full diffusion model from scratch, we believe that our training strategy could benefit the model even when training from scratch. Given that we are able to improve both generation and segmentation performance, we consider our method to provide training signals that align with the diffusion model’s knowledge. A recent work, REPA [5] trains diffusion networks from scratch without LoRA, while enforcing a feature‑alignment loss against DINO. This direct supervision accelerates convergence **and** improves generation fidelity, showing that alignment losses can be effective even when the entire model is updated.
> Nonetheless, REPA also shows that the layer selection for applying the alignment loss is crucial for both sample quality and training speed. Similarly, we expect layer-selection to still be of importance as it seems crucial to maintain alignment with the layer that naturally shows emergent properties, in terms of training efficiency, generation quality, and segmentation capabilities. We will add such discussions to the paper upon revision.
>
> [5] Yu, Sihyun, et al. "Representation alignment for generation: Training diffusion transformers is easier than you think." arXiv preprint arXiv:2410.06940 (2024).
>
> ### **Additional results on the perturbation experiment**
>
> To probe the importance of each cross‑attention block, we repeated the perturbation study but *fully masked* the image‑to‑text (I2T) channels one layer at a time. Qualitatively, removing I2T attention produces artifacts that mirror those seen when we previously applied a Gaussian blur to the same pathway, indicating comparable disruption mechanisms. To assess the effects in a quantitative manner, we adopted DreamBooth’s protocol [7] for generating 50 images per perturbation and reported the average change in DINO, CLIP‑I, and CLIP‑T scores relative to the unperturbed baseline. All three metrics dip most sharply when **layer 9** is masked, underscoring its pivotal role in aligning text prompts with visual content.
>
> | **Layer** | **L3** | **L7** | **L8** | **L9** | **L12** | **L13** | **L15** | **L16** |
> | --- | --- | --- | --- | --- | --- | --- | --- | --- |
> | DINO Score | 0.6933 | 0.6682 | 0.6816 | **0.4919** | 0.6665 | 0.6370 | 0.6779 | 0.6691 |
> | CLIP-I Score | 0.8898 | 0.8783 | 0.8772 | **0.8305** | 0.8737 | 0.8585 | 0.8830 | 0.8843 |
> | CLIP-T Score | 0.2912 | 0.2863 | 0.2844 | **0.2786** | 0.2930 | 0.2855 | 0.2908 | 0.2886 |
> | Harmonic mean | 0.4999 | 0.4895 | 0.4899 | **0.4394** | 0.4952 | 0.4809 | 0.4961 | 0.4925 |
>
> We thank the reviewer for the valuable suggestion, and we will include the qualitative and quantitative results as well as the discussions in the final version.
>
> [7] Ruiz, Nataniel, et al. "Dreambooth: Fine tuning text-to-image diffusion models for subject-driven generation." Proceedings of the IEEE/CVF conference on computer vision and pattern recognition. 2023.
>
> ### **Limitations of our method**
>
> We apologize for not including the limitations of our method in the supplementary material. To address this oversight, we provide the limitations below, which will certainly be included in our revised version.
>
> Our study deliberately focuses on dense perception and image‐generation tasks, leaving reasoning-centric evaluations (e.g., action recognition or spatial-relations QA) to future work because our supervision signal targets segmentation semantics rather than high-level reasoning. In addition, we attach the loss to a single “sweet-spot” layer per backbone for maximal simplicity; while this already delivers strong gains, a more exhaustive layer/timestep sweep—or the addition of auxiliary heads such as optical flow or pose—could yield incremental improvements without altering our core conclusions.

---

> > ### Comment · Reviewer_wi3R · 2025-08-05
> >
> > Thank you for you comments, my concerns are resolved.

---

> > > ### Author Response · Authors · 2025-08-05
> > >
> > > Thank you for your thoughtful review and for taking the time to engage with our work. We truly appreciate your response and feedback.

---

### Official Review · Reviewer_FeAF · 2025-07-03

**Clarity:** 3
**Significance:** 4
**Originality:** 3
**Rating:** 5
**Confidence:** 3

**Summary:**

This paper systematically analyze the attention structure of MM-DiT, especially Stable Diffusion 3, and reveal its implicit capabilities in zero-shot open vocabulary semantic segmentation. By decomposing the attention score distribution and the attention norm, identified a small number of attention heads that can consistently align text tokens with spatially coherent image regions, naturally producing high-quality zero shot segmentation masks. Based on this finding, they propose a lightweight LoRA based finetuning method to enhance the semantic grouping ability of these heads without degrading the image generation quality.

**Questions:**

see weakness, I'm happy to raise my rating if authors can solve my concerns.

**Ethical Concerns:**

["NO or VERY MINOR ethics concerns only"]

**Final Justification:**

The author's rebuttal solved my concers, and they have conducted a comprehensive experiments to show the advantages of their method. Their method is well designed and well contributed, I suggest to accept.

**Limitations:**

see weakness, mainly some experiments limitations.

**Paper Formatting Concerns:**

no formatting concerns.

**Quality:**

3

**Strengths And Weaknesses:**

## Strengths
1. indepth analysis on the attention mechanism, including the distrubution, L2 norms, and etc.. It is critical for identifying the property of the text to image alignment and image to text alignment.
2. the lora based method is simple but useful, it can improve the segmentation performance while maintain the generation quality, and have the capability of zero shot on multiple dataset.
3. This paper provide a strong theoretical support for the field of unified image generation and segmentation.

## Weaknesses
1. Only did experiments on SA-1B, in my opinion it is too detailed in some case, it would be better to provide the performance of the model trained on COCO.
2. ODISE is a strong model for the open vocabulary segmentation method that using diffusion model, there's no quantitative comparison provided for ODISE, and some more recent work (maybe consider SemFlow: Binding Semantic Segmentation and Image Synthesis via Rectified Flow)
3. line 215, you mentioned that you we fix the diffusion process at half of your 28 steps and apply the classifier free guidance, do you have any ablation of the choice of step 14 and step 28?

---

> ### Author Rebuttal · Authors · 2025-07-31
>
> ### **Generalizability to different models**
>
> To demonstrate that our insights are not tied to SD3, we replicated the entire analysis on two additional MM‑DiT variants—SD 3.5 [1] and Flux [2]. Summarizing the results, we observed consistent trends across all models (Note that the exact layer‑wise statistics can differ between models).
>
> Because the rebuttal cannot include external media, we present the key quantitative results here and will supply the corresponding plots and qualitative examples in the camera‑ready version. Following the methodology in the main paper, we use three complementary evaluation protocols:
>
> - **Attention‑norm analysis** – We compute the value‑vector norms for image, text, and actual prompt tokens at every layer, highlighting only the most informative layers for readability.
> - **Attention‑perturbation analysis** – For each layer, we perturb its cross‑attention and measure DINO, CLIP‑I, and CLIP‑T similarities between the original and perturbed images; larger drops indicate layers that are critical to image fidelity.
> - **Segmentation evaluation** – Using cross‑attention maps as pseudo‑masks, we evaluate mIoU on Pascal VOC to assess mask quality.
>
> In the table, bold indicates the highest entry for value norm and segmentation metrics, and the lowest for perturbation metrics.
>
> **Stable Diffusion 3.5 Medium**
>
> | Layer | 3 | 7 | 8 | 9 | 12 | 13 | 15 | 16 |
> | --- | --- | --- | --- | --- | --- | --- | --- | --- |
> | ***[Value norm ratio]*** |  |  |  |  |  |  |  |  |
> | Image tokens | **48.5** | 39.5 | 39.1 | 27.7 | 35.8 | 32.1 | 37.3 | 42.8 |
> | Text tokens | 17.4 | 29.3 | 29.8 | **41.5** | 27.4 | 41.2 | 33.3 | 23.2 |
> | Prompt tokens | 22.9 | 35.6 | 37.3 | **53.8** | 37.0 | 47.2 | 44.8 | 32.0 |
> | ***[Perturbation metric]*** |  |  |  |  |  |  |  |  |
> | DINO | 0.8910 | 0.7814 | 0.8314 | **0.6011** | 0.8432 | 0.7929 | 0.8507 | 0.9056 |
> | CLIP-I | 0.9511 | 0.9232 | 0.9333 | **0.8701** | 0.9407 | 0.9319 | 0.9434 | 0.9599 |
> | CLIP-T | 0.3188 | 0.3194 | 0.3181 | **0.3108** | 0.3182 | 0.3177 | 0.3151 | 0.3175 |
> | Harmonic mean | 0.5649 | 0.5460 | 0.5537 | **0.4974** | 0.5564 | 0.5472 | 0.5546 | 0.5665 |
> | ***[Semseg eval]*** |  |  |  |  |  |  |  |  |
> | PascalVOC | 72.6 | 82.8 | 82.8 | **84.7** | 83.4 | 82.9 | 78.3 | 75.3 |
>
> **Flux.1-dev**
>
> | Layer | 0 | 2 | 6 | 8 | 9 | 10 | 12 | 15 | 17 | 26 |
> | --- | --- | --- | --- | --- | --- | --- | --- | --- | --- | --- |
> | ***[Value norm]*** |  |  |  |  |  |  |  |  |  |  |
> | Image tokens | 57.1 | 85.8 | 41.9 | 54.8 | 52.8 | 53.6 | 56.7 | 88.9 | **89.3** | 70.6 |
> | Text tokens | 63.4 | **91.9** | 69.8 | 70.8 | 67.9 | 74.1 | 66.3 | 58.0 | 70.1 | 56.7 |
> | Prompt tokens | 72.4 | 106.3 | 104.4 | 115.1 | 103.8 | 120.6 | 127.1 | 108.6 | **139.8** | 57.3 |
> | ***[Perturbation metric]*** |  |  |  |  |  |  |  |  |  |  |
> | DINO | 0.9737 | 0.9922 | 0.9947 | 0.9829 | 0.9780 | 0.9635 | **0.9524** | 0.9820 | 0.9617 | 0.9699 |
> | CLIP-I | 0.9899 | 0.9972 | 0.9978 | 0.9938 | 0.9906 | 0.9881 | **0.9832** | 0.9946 | 0.9896 | 0.9920 |
> | CLIP-T | 0.3022 | 0.3038 | 0.3029 | 0.3031 | 0.3039 | 0.3040 | **0.3013** | 0.3036 | 0.3044 | 0.3033 |
> | Harmonic mean | 0.5611 | 0.5657 | 0.5650 | 0.5636 | 0.5636 | 0.5618 | 0.5569 | 0.5641 | **0.5546** | 0.5622 |
> | ***[Semseg eval]*** |  |  |  |  |  |  |  |  |  |  |
> | PascalVOC | 65.9 | 68.0 | 70.0 | 73.8 | 73.3 | 71.6 | 79.7 | 74.5 | **83.5** | 77.0 |
>
> From the table above, we can similarly observe the correlation between value norm and segmentation performance among layers for both Flux and Stable Diffusion 3.5. While SD3.5 appears to have layer 9, identical to SD3, to exhibit strong semantic grounding, Flux shows layer 12 and 17 to have a similar tendency. This hints that our observation and methodology are not proprietary for SD3, but can be applied to other DiT-based diffusion models with multi-modal attention, highlighting the generalizability of our insights and findings.
>
> ### **Ablation on the choice of timestep**
>
> Initially, we chose **t=14** as our early experiments on ADE20K showed the best result for t=14 when we conducted a coarse search over the timesteps. To further explore the effects of timesteps, we evaluate over **all** 28 timesteps in VOC and COCO-Object, and present the results below. Note that we only show some of the best and worst performing layers for brevity.
>
> | **Dataset \ Timestep** | **1** | **2** | **6** | **7** | **8** | **14** | **21** | **28** |
> | --- | --- | --- | --- | --- | --- | --- | --- | --- |
> | **Pascal VOC** | 73.9 | 84.5 | **89.2** | 89.0 | **89.2** | 87.8 | 81.0 | 70.5 |
> | **COCO-Object** | 43.9 | 55.4 | 62.0 | **62.1** | **62.1** | 58.2 | 48.5 | 25.4 |
>
> The results reveal that t=8, in fact, consistently outperforms our original choice of t=14 for both VOC and COCO-Object, with t=6 and 7 similarly showing strong results. While we default the timestep to t=14 for additional experiments presented in the rebuttals to avoid confusion, we thank the reviewers for pointing this out and promise to update the scores as well as include the ablation on timesteps in the revised manuscript.
>
>
> ### **Comparison with COCO-trained model**
>
> We agree that the masks predicted by SAM can be overly fine‑grained, which can hinder training by enforcing the attention maps within the diffusion model to very small regions. In this regard, we carry out the same training with COCO images and masks, and report the results in weakly and unsupervised settings, as well as the generation performance with Clipscore below.
>
> |  |  | **Weakly** |  | **Unsupervised** |  | **Generation** |
> | --- | --- | --- | --- | --- | --- | --- |
> | Model | Dataset | VOC | COCO-Obj | Cityscapes | COCO-Stuff-27 | Clipscore |
> | Ours (Zero-shot) | - | 86.4 | 58.0 | 24.2 | 49.7 | 27.1437 |
> | Ours (Trained) | SA-1B | 87.0 | 58.0 | 24.2 | 50.8 | 27.2447 |
> | Ours (Trained) | COCO | **88.7** | **59.2** | **24.4** | **53.5** | **27.2760** |
>
> We can observe that not only both weakly and unsupervised performance showed improvements, but also the generation performance has improved when compared to being trained on SA-1B. This suggests the importance of mask granularity in our fine‑tuning framework. We appreciate the reviewer’s constructive suggestion and will incorporate the COCO‑trained model into the manuscript upon revision.
>
> ### **Comparison with ODISE and SemFlow**
>
> | **Method** | **Dataset** | **Supervision** | **VOC20** | **Object** | **PC59** | **ADE** | **City** |
> | --- | --- | --- | --- | --- | --- | --- | --- |
> | ***[Closed-set]*** |  |  |  |  |  |  |  |
> | SemFlow | COCO-Stuff | Label + mask | N/A | N/A | N/A | N/A | N/A |
> | ***[Fully-supervised]*** |  |  |  |  |  |  |  |
> | ODISE | COCO-Panoptic | Label + mask | - | - | 57.3 | 29.9 | - |
> | ***[Weakly-supervised/ Zero-shot]*** |  |  |  |  |  |  |  |
> | Ours (Zero-shot) | - | *None* | 86.4 | 58.0 | 46.1 | 31.9 | 19.0 |
> | Ours (Trained) | SA-1B | mask | 87.0 | 58.0 | 46.3 | 32.0 | 19.2 |
> | Ours (Trained) | COCO | mask | 88.7 | 59.2 | 48.2 | 32.2 | 19.7 |
>
> We appreciate the reviewer for suggesting ODISE [1] and SemFlow [2] for comparison. While both models are based on diffusion models, we notice that both ODISE and SemFlow are trained with different levels of supervision and evaluated in different settings, which prevents a fair comparison to our work. SemFlow is trained in a closed-set manner, which cannot perform open-vocabulary segmentation, and ODISE requires full supervision of having class-labeled masks. By contrast, our model operates in a *weakly supervised, zero-shot* setting and—without any class labels, while lightweight fine-tuning with inexpensive SA-1B or COCO masks pushes the scores. Nonetheless, we show comparisons in the table above as a reference, and we will add clarifications regarding the different settings of different methods to the manuscript.
>
> [1] Xu, Jiarui, et al. "Open-vocabulary panoptic segmentation with text-to-image diffusion models." Proceedings of the IEEE/CVF conference on computer vision and pattern recognition. 2023.
>
> [2] Wang, Chaoyang, et al. "Semflow: Binding semantic segmentation and image synthesis via rectified flow." Advances in Neural Information Processing Systems 37 (2024): 138981-139001.
>
>
> ### **Limitations of our method**
>
> We apologize for not including the limitations of our method in the supplementary material. To address this oversight, we provide the limitations below, which will certainly be included in our revised version.
>
> Our study deliberately focuses on dense perception and image generation tasks, leaving reasoning-centric evaluations (e.g., action recognition or spatial-relations QA) to future work because our supervision signal targets segmentation semantics rather than high-level reasoning. In addition, we attach the loss to a single “sweet-spot” layer per backbone for maximal simplicity; while this already delivers strong gains, a more exhaustive layer/timestep sweep—or the addition of auxiliary heads such as optical flow or pose—could yield incremental improvements without altering our core conclusions.

---

> > ### Comment · Reviewer_FeAF · 2025-08-01
> >
> > Thanks the detailed and impressive reply, the authors solved my concern.

---

> > > ### Author Response · Authors · 2025-08-05
> > >
> > > We’re glad to hear that our response has addressed all the concerns. If there are no remaining issues, we would greatly appreciate it if the reviewer could consider raising the score. Thank you again for your thoughtful and constructive feedback.

---

> > > > ### Comment · Reviewer_FeAF · 2025-08-05
> > > >
> > > > Thanks again for your very detailed experiments, I have raised my score to accept in the final justification.

---

### Note · Authors · 2025-08-12

Dear reviewers, ACs and SACs,

We thank the reviewers for their thoughtful and constructive feedback, and the Area Chair for thoroughly reviewing our work.

The discussion allowed us to clarify our contributions, present additional evidence, and further strengthen the paper. Key improvements include demonstrating applicability to other MM-DiT models (Flux, SD3.5), providing additional ablations and comparisons, and exploring training with masks of varying granularity, which we find beneficial to our framework.

We are pleased that all reviewers acknowledged our responses and confirmed that most of their concerns have been addressed. We will incorporate these discussions into the revised version, and we strongly believe that it would improve the clarity and solidity of our work greatly.

Best regards, Authors

---

### Decision · Program_Chairs · 2025-09-17

**Decision:**

Accept (poster)

**Comment:**

This paper analyses the structure of attention maps of MM-DiTs, to study the alignment between semantic information in image and text tokens. Following this analysis, the paper proposes a lightweight LoRA-based fine-tuning to improve controlability by addressing the relevant attention heads. The proposed approach is based on an interesting analysis and yields results comparable to respective baselines, while improving alignment.

The reviews agree that the analysis is valuable and can help designing better and more efficient MM-DiTs. Initial criticism on the method limitations, and limited evaluation, for example only on SD3, has been discussed and the authors have provided significant additional analysis and results in the rebuttal. Therefore, after the discussion, all reviewers agree that major concerns have been addressed and recommend the paper for acceptance. The AC agrees.